# Multiple temperatures and melting of a colloidal active crystal

Helena Massana-Cid [1]✉, Claudio Maggi [1,3]✉, Nicoletta Gnan[1,4], Giacomo Frangipane [1,2] & Roberto Di Leonardo [1,2]

Thermal fluctuations constantly excite all relaxation modes in an equilibrium crystal. As the temperature rises, these fluctuations promote the formation of defects and eventually melting. In active solids, the self-propulsion of "atomic" units provides an additional source of non-equilibrium fluctuations whose effect on the melting scenario is still largely unexplored. Here we show that when a colloidal crystal is activated by a bath of swimming bacteria, solvent temperature and active temperature cooperate to define dynamic and thermodynamic properties. Our system consists of repulsive paramagnetic particles confined in two dimensions and immersed in a bath of light-driven *E. coli*. The relative balance between fluctuations and interactions can be adjusted in two ways: by changing the strength of the magnetic field and by tuning activity with light. When the persistence time of active fluctuations is short, a single effective temperature controls both the amplitudes of relaxation modes and the melting transition. For more persistent active noise, energy equipartition is broken and multiple temperatures emerge, whereas melting occurs before the Lindemann parameter reaches its equilibrium critical value. We show that this phenomenology is fully confirmed by numerical simulations and framed within a minimal model of a single active particle in a periodic potential.

Colloidal systems have been successfully studied as model atomic solids thanks to their experimentally accessible length and time scales, along with their known and controllable interactions. They have been used to elucidate very debated issues connected to the nature of melting transition in two dimensions[1], the dynamical arrest in glass transition[2], and the vibrational excitations in disordered solids[3]. A new class of solid materials, known as active solids[4–6], has recently attracted increasing attention. These materials consist of active, self-propelled particles embedded in an elastically coupled network. This mix generates a wide variety of macroscopic collective motions[7,8] that have no counterpart in systems at equilibrium while they are often found in living systems[9]. Nevertheless, on a more "atomistic" and fundamental level, there is limited experimental evidence on how activity alters the conventional properties of solids in equilibrium such as the equipartition of energy among relaxation modes[10] and the microscopic origin of melting[11]. Whereas some numerical studies have shown that the two-step melting scenario of 2d crystals is qualitatively preserved when activity is introduced[12–15], more recent simulation work has pointed out that two different effective temperatures control the large-scale elastic deformations of the crystal structure and the small-scale bond-order fluctuations[16,17]. The primary obstacle to tackle these issues experimentally is the necessity for a system that is simultaneously active and possesses a precisely defined, controllable geometry, and interactions. While self-propelling synthetic active particles can self-organize into dynamic crystals when colliding[18–20], or into polycrystals when sedimenting[21,22], particle velocities and fluctuations are hard to access in these close-packed systems[23–25]. Moreover, the

[1]Dipartimento di Fisica, Sapienza Università di Roma, Piazzale A. Moro 5, 00185 Rome, Italy. [2]NANOTEC-CNR, Soft and Living Matter Laboratory, Institute of Nanotechnology, Piazzale A. Moro 5, 00185 Rome, Italy. [3]Present address: NANOTEC-CNR, Soft and Living Matter Laboratory, Institute of Nanotechnology, Piazzale A. Moro 5, 00185 Rome, Italy. [4]Present address: CNR Institute of Complex Systems, Uos Sapienza, Piazzale A. Moro 5, 00185 Rome, Italy. ✉e-mail: helena.massanacid@uniroma1.it; claudio.maggi@cnr.it

self-propulsion mechanism can become ineffective or perturbed by the presence of close neighbors.

Here we show an experimental realisation of a large ordered and loose-packed colloidal active solid with interactions and activity tunable in-situ and on-command. Our system consists of a two-dimensional crystal composed of self-ordered magnetic particles activated[26–29] by a photokinetic bacterial bath. This non-equilibrium solid is excited by two fluctuating forces: the thermal noise due to the solvent and the stochastic interactions with swimming bacteria. The strength of these two components can be adjusted dynamically by tuning inter-particle interactions through the magnetic field and by changing the intensity of the green light that powers bacteria. We study the crystal's harmonic behavior when excited by these two forces and find that they act differently on the system's degrees of freedom, resulting in the coexistence of multiple temperatures and revealing the non-equilibrium nature of active lattice fluctuations. Furthermore, these fluctuations can break bonds and generate topological defects in our system, allowing us to explore a novel melting route by increasing particle activity. We find that in active melting, at large scales, the system appears to qualitatively follow the two-step phase transition predicted by KTHNY theory[30]. In contrast, at a more local level, persistent active forces promote particle hopping that triggers melting when the nearest neighbor fluctuations, measured by the Lindemann parameter, are much smaller than what is observed in equilibrium[31]. Overall, our system will serve as an unprecedented experimental test-bed for the study of many effects that have so far remained largely theoretical predictions.

## Results

We assemble the active solid by applying a magnetic field **H** to a sample consisting of two-dimensionally confined paramagnetic colloids of 4.5 μm diameter immersed in a bath of photokinetic bacteria (see Methods). When the applied magnetic field is perpendicular to the sample plane, there is an isotropic repulsion energy between particles separated by a distance $a$ of $U_M = \mu_0\chi^2 H^2/4\pi a^3$, where $\chi$ is the particle's magnetic susceptibility and $\mu_0$ is the magnetic permeability. This repulsion maximizes the inter-particle distance, inducing the formation of a triangular lattice of non-close packed repulsive particles (Fig. 1a,b, Supplementary Movie 1).

On the other hand, the bacteria in the bath induce active motion into the crystal by pushing its particles[26]. These cells are *E. coli*[32] expressing the light-driven proton pump proteorhodopsin (see

Methods). The sample is sealed so that, after the cells have consumed all the oxygen in the buffer through respiration, the proton motive force drops down and the flagellar motors stop spinning and start responding to external green light stimuli. Using a green LED, we can control swimming speed and thus the induced activity by illuminating the sample with green light of different intensity $I$. When there is no green light applied, bacteria do not move and the system consists of confined Brownian particles with a diffusion coefficient $D_T$. When we apply green light $I \neq 0$, the bacteria swim and push the particles so the amplitude of their fluctuations around their lattice position increases (Fig. 1c, Supplementary Movie 2). This results in colloids acquiring a persistent motion with a characteristic time $\tau$ and an effective active diffusion coefficient $D_A$. The effect of the active bath on the colloids, influenced by factors such as bacteria concentration and light intensity $I$ (speed), can be characterized by $D_A$ and $\tau$[26]. We calibrate these parameters for each experiment, in the absence of a magnetic field for various green light intensities (see Methods, Supplementary Fig. 1).

In the limit of short persistence time $\tau$, active particles with mobility $\mu$ behave like "hotter" equilibrium systems with a higher thermal energy $(D_A + D_T)/\mu$ obtained from a generalized Stokes-Einstein relation[33]. Dividing this by the magnetic interaction energy scale $U_M$ we can introduce an adimensional global temperature $T^*$:

$$T^* = \frac{4\pi}{\mu_0\chi^2}\frac{a^3}{H^2}\frac{D_T + D_A}{\mu} \tag{1}$$

By independently tuning the external magnetic field and the intensity of the green light we can control this global effective temperature and at the same time the relative contributions from the solvent and the active bath.

One of the most intriguing questions that follows from this picture is whether there is any qualitative difference between active and thermal excitations in the solid and as a result a violation of the classic equipartition theorem. It has been predicted[34], that the equipartition theorem can be broken in out-of-equilibrium systems such as those constituted by active particles, if the external potential introduces time-scales comparable with the active motion's persistence time. However, this has never been observed in a real system. To understand how activity affects fluctuations in the active crystal we study its normal mode band structure[10,35]. We must importantly note that here we

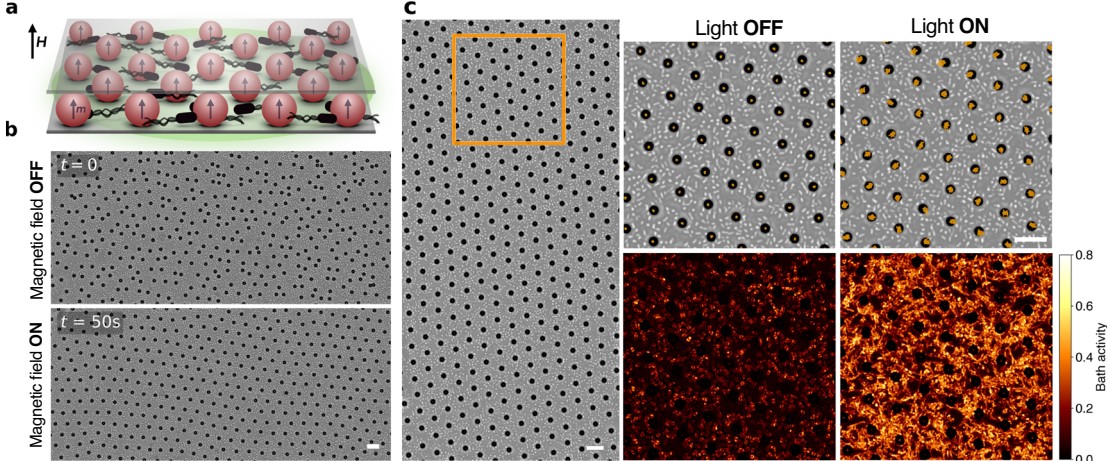

**Fig. 1 | A colloidal active crystal powered by bacteria. a** Side-view schematic of the experimental system: 2D confined magnetic particles with moment **m** aligned with an external field **H** forming a triangular lattice in a bacterial bath powered by green light. **b** Microscope snapshots of the assembly of the active magnetic solid in a bacterial bath, before and after applying a magnetic field (Supplementary Movie 1). **c** Microscope snapshot of an assembled magnetic solid in a bacterial bath after equilibration (Supplementary Movie 2). Corresponding trajectories of the passive (Light OFF) and active crystal (Light ON) in the enlarged area of the highlighted box in orange (top). Bath activity, specifically calculated using the relative variance of bacteria local density (bottom). All scale bars are 20 μm.

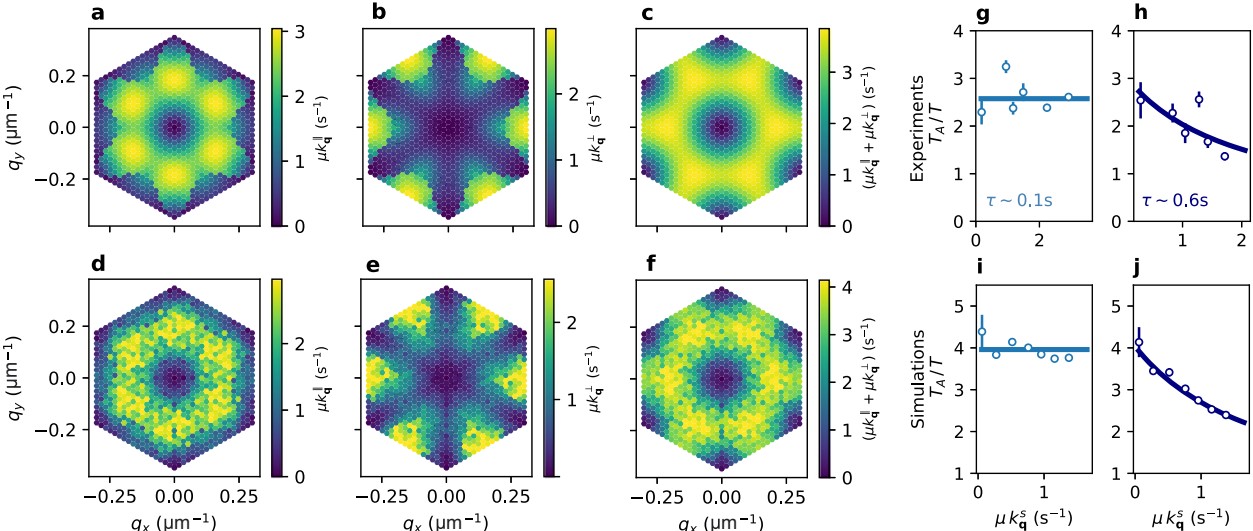

**Fig. 2 | Coexistence of multiple temperatures in the active crystal.** Map in the reciprocal space $(q_x, q_y)$ of the **a** longitudinal $k_{\mathbf{q}}^{\parallel}$ and **b** transverse $k_{\mathbf{q}}^{\perp}$ eigenvalues of the dynamical matrix and **c** their sum obtained numerically for a dipolar two dimensional crystal. Map in the reciprocal space $(q_x, q_y)$ of the **d** longitudinal $k_{\mathbf{q}}^{\parallel}$ and **e** transverse $k_{\mathbf{q}}^{\perp}$ eigenvalues and **f** their sum obtained in the experiment for the passive crystal. **g** Relative active temperature $T_A/T$ as a function of the eigenvalues

$k_{\mathbf{q}}^s$ for experiments at short persistence time $\tau = 0.1\,\mathrm{s}$ (points) and theoretical fit (continuous line). Error bars correspond to the standard deviation. Same quantities are reported in **h** for experiments at $\tau = 0.6\,\mathrm{s}$. **i, j** report same plots in simulations of AOUPs. For large $\tau$ (**h, j**), the active data are well fitted by a hyperbola according to the theory for AOUPs. For low $\tau$ (**g, i**), the quantity $T_A/T$ is nearly constant.

are not discussing phonon vibrational modes. Since the crystal is immersed in a viscous fluid, the colloids' motion is over-damped, and the trajectory of a particle is composed of a superposition of normal relaxation modes[36]. Deep in the crystalline phase particles undergo small displacements with respect to their equilibrium positions that we denote by $\mathbf{u}_i = \mathbf{r}_i - \langle \mathbf{r}_i \rangle$. Here $\mathbf{r}_i$ and $\langle \mathbf{r}_i \rangle$ are, respectively, the instantaneous position and the (time averaged) equilibrium position of the $i$-th particle. We then consider the Fourier components $\mathbf{u}_{\mathbf{q}}$ of the displacement field $\mathbf{u}_i$ with $\mathbf{q}$ a vector in the reciprocal lattice defined by the equilibrium positions $\langle \mathbf{r}_i \rangle$. Introducing the Fourier-transformed dynamical matrix[35] $\mathbf{D}_{\mathbf{q}}$ (see Methods), the average harmonic potential energy of the crystal can be thus written as:

$$U = \frac{1}{2} \sum_{\mathbf{q}, \alpha, \beta} D_{\mathbf{q}}^{\alpha, \beta} \langle u_{\mathbf{q}}^{\alpha} u_{-\mathbf{q}}^{\beta} \rangle \qquad (2)$$

When displacements are represented in longitudinal and transverse coordinates $u_{\mathbf{q}}^{\parallel}, u_{\mathbf{q}}^{\perp}$ the dynamical matrix $\mathbf{D}_{\mathbf{q}}$ is diagonal with eigenvalues $k_{\mathbf{q}}^{\parallel}, k_{\mathbf{q}}^{\perp}$ so that the total potential energy is a sum of quadratic terms and, if in equilibrium, energy equipartition imposes that:

$$k_{\mathbf{q}}^s p_{\mathbf{q}}^s = k_B T \qquad (3)$$

with $p_{\mathbf{q}}^s = \langle |u_{\mathbf{q}}^s|^2 \rangle$ the mean squared amplitude of displacement fluctuations with wavevector $\mathbf{q}$ and polarization $s = \parallel, \perp$. In other words, when dynamics is only driven by thermal forces, every mode provides an independent and consistent measurement of the equilibrium temperature through Eq. (3). In active systems, non-equilibrium fluctuations may give rise to strong deviations from Boltzmann statistics when the persistence time of the active noise competes with other internal relaxation times. When the active noise is Gaussian, i.e. we consider Active Ornstein-Uhlenbeck Particles (AOUPs), the expression for the mean potential energy of a harmonic oscillator becomes a simple generalization of the equipartition formula[34]. This result can be generalized to a harmonic crystal for which it becomes[37]:

$$k_{\mathbf{q}}^s p_{\mathbf{q}}^s = k_B \left[ T + T_A \left( k_{\mathbf{q}}^s \right) \right] \qquad (4)$$

with $T_A(k_{\mathbf{q}}^s)$ a mode-specific effective active temperature given by

$$T_A(k_{\mathbf{q}}^s) = \frac{D_A}{\mu k_B} \frac{1}{1 + \mu k_{\mathbf{q}}^s \tau} \qquad (5)$$

A similar notion of effective temperatures has been introduced for glassy systems[38], where multiple temperatures can be measured by "thermometers" that probe the system's dynamics on different time-scales. In the present case, the relaxation modes in the crystal play the role of thermometers that measure different effective temperatures, revealing the non-equilibrium nature of the active system. Specifically, stiffer modes that relax on a characteristic time $1/(\mu k_{\mathbf{q}}^s)$ shorter than the persistence time of the active noise $\tau$, will be "colder" relative to softer modes such as those at longer wavelengths. This coexistence of multiple effective temperatures has been theoretically predicted but never observed in real active systems. To find evidence for this effective temperature spectrum we need to measure the mode-specific potential energy $k_{\mathbf{q}}^s p_{\mathbf{q}}^s$. Mode stiffnesses $k_{\mathbf{q}}^s$ can be obtained by measuring mode amplitudes $p_{\mathbf{q}}^s$ in a passive system where green light is off and bacteria are non-motile. In Fig. 2d–f we report experimental $k_{\mathbf{q}}^s$ as a density map on the first Brillouin zone. Then we turn on activity compute the mean squared amplitudes of relaxation modes $p_{\mathbf{q}}^s$ and obtain the effective active temperatures as defined by Eq. (4):

$$\frac{T_A(k_{\mathbf{q}}^s)}{T} = \frac{k_{\mathbf{q}}^s p_{\mathbf{q}}^s}{k_B T} - 1 \qquad (6)$$

Using Eq. (6) to extract $T_A$ we test Eq. (5) in Fig. 2g for an experimental active crystal where $\tau = 0.1\,\mathrm{s}$. To reduce noise, we average all $p_{\mathbf{q}}^{\parallel, \perp}$ whose corresponding $k_{\mathbf{q}}^s$ fall in a given interval. The ratio $T_A/T$ in Fig. 2g shows no systematic deviations from the horizontal line. In other words, when $\tau$ is small, all modes probe the same effective temperature. In contrast, when we tune light intensity to maximize $\tau$, we find a mode specific temperature that decreases by a factor two with increasing relaxation rates $\mu k_{\mathbf{q}}^s$. This decay is consistent with the prediction in Eq. (5) shown by the solid line. By fitting the data with $\tau$ and $D_A/D_T$ as free parameters, we obtain $\tau = 0.5\,\mathrm{s}$ and $(D_A/D_T) = 3$, in reasonable agreement with independent measurements of the same constants in the

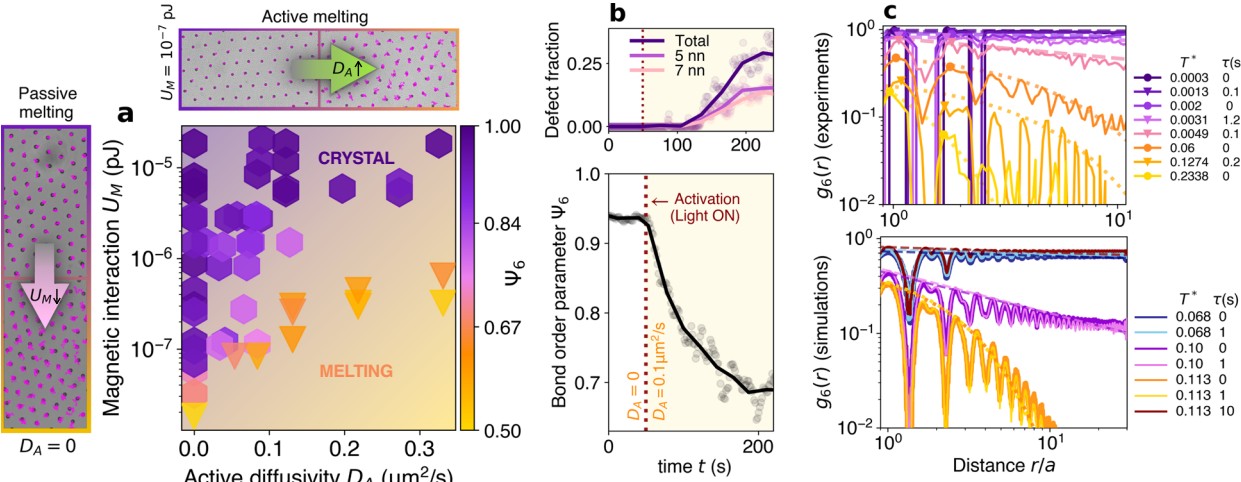

**Fig. 3 | Out-of-equilibrium phase diagram and active melting route. a** Phase diagram as a function on the active diffusion $D_A$ and magnetic interaction $U_M$. Hexagonal symbols correspond to the crystalline phase, and triangles to melting systems, according to the Lindemann criterion. Each point is colored according to the structural order (average bond order parameter $\Psi_6$). Snapshots and particle trajectories for different systems around the phase diagram, and effect of changing $D_A$ and $U_M$: passive melting when decreasing the magnetic field without activity (Supplementary Movie 3), active melting when increasing light intensity and consequently $D_A$ at constant magnetic field and $U_M$ (Supplementary Movie 4). **b** Evolution of the bond order parameter $\Psi_6$ of a crystal when light is suddenly turned on at $t = 50$ s so that it is activated at $D_A = 0.1\,\mu m^2/s$. In this experiment $U_M = 10^{-7}$ pJ and $\tau = 0.6$ s. Top graph shows the evolution on the defect fraction in the same system, of defects consisting of particles with 5 nearest neighbors, 7 nearest neighbors, and the total. **c** Orientational correlation function for different global temperatures $T^*$ (Eq. (1)). with fits to power law (dashed lines) and exponential decay (dotted lines).Experimental results (top). Simulation results for groups of active and passive systems with identical total temperature (bottom).

absence of a magnetic field ($\tau = 0.6$ s and $D_A/D_T = 2.5$, see Methods). To check the robustness of our method, we perform the exact same analysis for numerical simulations of a crystal composed of AOUPs[39] interacting via dipolar repulsive forces in two dimensions as shown in Fig. 2i, j. For these simulations, we choose all dynamic parameters close to the experimental ones. Simulations display a very similar pattern of mode-specific temperatures when $\tau$ becomes comparable to the characteristic relaxation times of harmonic modes.

The harmonic analysis described above was performed on "cold" crystals with global temperature values $T^* < 0.01$. We have two routes to explore higher temperature phases. The first one is by decreasing the magnetic interaction energy $U_M$, as already explored in passive systems[11] (see also Supplementary Movie 3). The second is by tuning activity $D_A$ and melting the system through non-equilibrium fluctuations (Supplementary Movie 4). Along this entirely new route, we can explore how equilibrium and off-equilibrium temperatures cooperate to define the phases of the system. We characterise the system's structural states at different $D_A$ and $U_M$ using the orientational bond order parameter:

$$\Psi_6 = \frac{1}{N}\sum_{i=1}^{N}|\Psi(\mathbf{r}_i)| \tag{7}$$

where $\Psi(\mathbf{r}_i) = \sum_{j=1}^{N_i} e^{i6\theta_{ij}}/N_i$, $N_i$ is the number of neighboring particles, and $\theta_{ij}$ the angle between a fixed axis and the bond joining particles $i$ and $j$. The parameter $\Psi_6$ quantifies the hexagonal symmetry in the crystal lattice, achieving a maximum value of 1 when each particle is surrounded by six symmetrically arranged nearest neighbors. We then use the dynamic Lindemann parameter as melting criterion[11]:

$$\gamma_L(t) = \frac{\langle(\Delta\mathbf{r}_i(t) - \Delta\mathbf{r}_j(t))^2\rangle}{2a^2} \tag{8}$$

where $\Delta\mathbf{r}_i$ is the $i$th particle displacement after a time $t$: $\Delta\mathbf{r}_i(t) = \mathbf{r}_i(t) - \mathbf{r}_i(0)$, $i$ and $j$ are nearest neighbors and $a$ is the interparticle distance. If $\gamma_L(t)$ saturates for long $t$ it means that the neighbors are coupled and the system is a solid. On the other hand, when $\gamma_L$

diverges, it means the neighbors diffuse away from each other and the system melts. Thus, with $\Psi_6$ to quantify the sixfold orientational symmetry and the dynamic Lindemann parameter as a melting criterion we draw a phase diagram depending on $U_M$ and $D_A$ (Fig. 3a). Crystalline phases are found in the top left corner corresponding to low activity or high magnetic interaction while melting occurs when moving either to lower $U_M$ or higher $D_A$. This novel melting by activity can be suddenly triggered just by turning on green light which results in a fast and progressive decrease of orientational order (Fig. 3b). After a couple of minutes $\Psi_6$ decays from 0.94 to 0.67 while defects, i.e. particles with 5 or 7 nearest neighbors, begin to appear reaching a total fraction 30%.

We showed above through mode analysis that short wavelength modes may have a lower effective temperature if their relaxation time is comparable to $\tau$. This could affect the activated hopping of particles and inhibit melting. Therefore, we check whether the melting structural transition is solely predicted by $T^*$ or if it is influenced by $\tau$. For that, we calculate the orientational correlation function $g_6(r)$:

$$g_6(r) = g_6(|\mathbf{r}_i - \mathbf{r}_k|) = \langle\Psi(\mathbf{r}_i)\,\Psi^*(\mathbf{r}_k)\rangle \tag{9}$$

In equilibrium, 2D colloidal crystals melt in two steps according to KTHNY theory[1]: firstly from crystalline or long-range orientational order ($g_6(r) \sim 1$) to hexatic or quasi-long-range order ($g_6(r) \sim r^{-\eta_6}$), and then to short-range order or isotropic phase ($g_6(r) \sim e^{-r/\xi_6}$). When melting occurs mainly via active fluctuations, we do not observe any discernible discrepancies with KTHNY: by increasing $T^*$ the system transitions from crystalline to hexatic and then to isotropic, both in the case of $D_A = 0$ and $D_A \neq 0$ (Fig. 3c). Furthermore, if we plot the bond order parameter $\Psi_6$ all data of the phase diagram as a function of $T^*$ we see that all points fall approximately on the same curve (see Supplementary Fig. 2). To further validate this assertion and access larger systems and larger persistence times $\tau$, we perform numerical simulations on active and passive systems characterized by exactly the same $T^*$. Our findings reveal that for the experimentally accessible parameters of $\tau \sim 1$ s, the orientational correlation remains nearly indistinguishable between the two, except for a slight increase in order

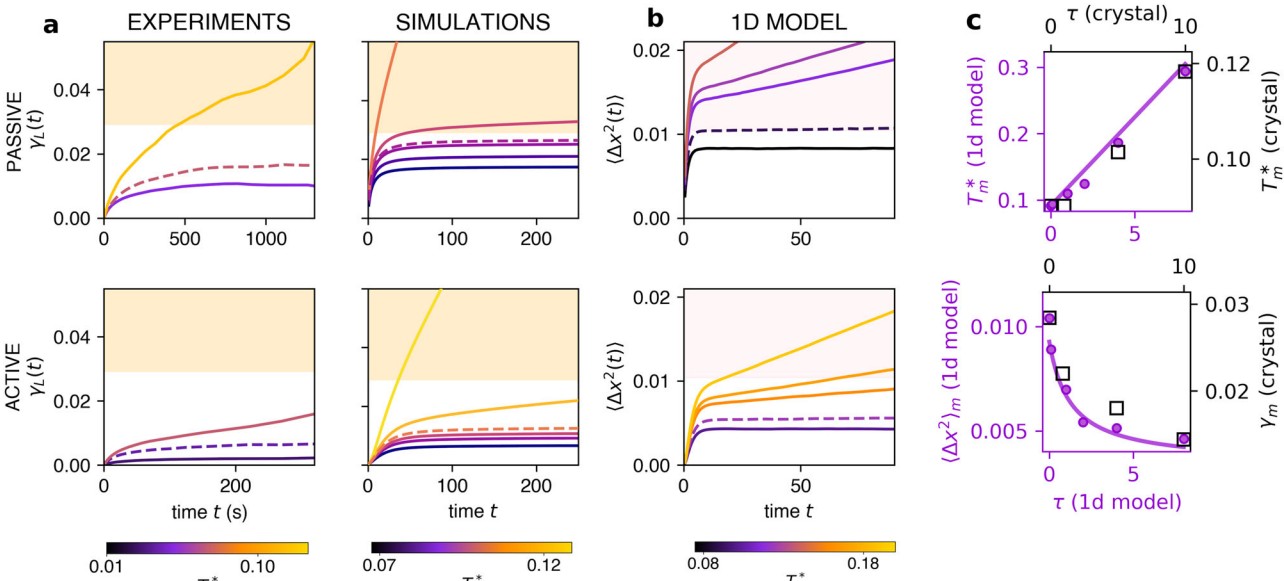

**Fig. 4 | Active melting: anomalous Lindemann parameter and increase of the melting temperature. a** Lindemann parameter $\gamma_L(t)$ in experiments and simulations of passive (1st row) and active (2nd row) melting. The shaded orange area corresponds to values over the critical threshold of Lindemann measured in the passive case for the simulations $\gamma_L^* = 0.03$. **b** Mean squared displacement $\langle \Delta x^2(t) \rangle$ of an active particle trapped in a sinusoidal potential. The shaded area to the critical threshold of the squared displacement in the passive case. All curves in **a**, **b** are colored according to the total effective temperature $T^*$. Saturated $\gamma_L(t)$ corresponds to solid-state systems while diverging $\gamma_L(t)$ to melting. Dashed lines in are the last measurement before melting. **c** Melting temperature $T_m^*$ in both simulations and 1d model as a function of persistence time $\tau$ (top). Maximum plateau before melting of the Lindemann parameter for the simulated crystal ($\gamma_m$) and of the mean-squared-displacement for the 1d model $\langle \Delta x^2 \rangle_m$ (bottom). The solid line corresponds to the theoretical prediction (see Methods).

in the active case. However, for a larger persistence time $\tau \sim 10$ s, the system is still crystalline at a temperature that melts the passive counterpart, revealing a hotter melting temperature $T_m^*$ for the active case. Furthermore, we still observe the 2-step melting scenario for larger $\tau$ and larger systems (Supplementary Figs. 3–4).

We have shown through harmonic analysis that the short wavelength modes are particularly affected by active noise, exhibiting decreased fluctuations as though they were in equilibrium at a lower effective temperature compared to the longer wavelengths. This observation suggests that the peculiar aspects of active melting may emerge in measurements of local quantities, such as the Lindemann parameter which captures fluctuations between neighbors. It was measured in a similar system in equilibrium conditions[11] a comprehensive Lindemann melting threshold of $\gamma_m = 0.033$. Beyond this threshold, i.e. when the system fluctuates such that $\gamma(t) > \gamma_m$, the neighbors start to diffuse away from each other and the system melts. We explore this scenario both in experiments and simulations. In the latter, similarly to ref. [11], we find a critical melting value of $\gamma_m = 0.029$ in the equilibrium system. Remarkably, in active systems, we see that this criterion breaks down and $\gamma_L(t)$ starts diverging at lower critical values (Fig. 4a). This indicates that the nearest neighbor bond fluctuations that the crystal has just before melting are hindered in the active case compared to those of a system in equilibrium. We see that in the experiments active melting occurs at lower plateaus for the experimental persistence time (0.1-1s) than in simulations (1-10s). This difference may be caused by the latter not capturing the longer tails of the displacement probabilities[40] $P(\Delta x)$ observed experimentally, which could potentially trigger earlier melting. Specifically, we observe wider tails in the distributions when the active diffusion $D_A$ increases (Supplementary Fig. 5). In simulations we also find that for highly active systems, i.e. with large $\tau$, the melting effective temperature $T_m^*$ increases, as we saw in Fig. 3c, even though the resulting maximum plateau of Lindemann $\gamma_m$ decreases. To understand this counterintuitive effect, we propose a simple schematic model: i.e. we consider one single AOUP particle moving over a one-dimensional sinusoidal

potential (see Methods). At low effective temperatures the particle is trapped in a local minimum, in analogy with particles being trapped by the nearest neighbor cage. The mean squared displacement $\langle \Delta x^2(t) \rangle$ in this 1d-model is the analog of the dynamic Lindemann parameter $\gamma_L(t)$ in the crystal. As we increase the noise strength the particle begins hopping between minima (which is the analogue of melting) as shown in Fig. 4b. Also, in this case, we observe that the plateau value of $\langle \Delta x^2(t) \rangle$ at the transition (denoted by $\langle \Delta x^2 \rangle_m$) decreases sensibly upon increasing $\tau$ and that the effective temperature needs to be higher to induce melting at longer $\tau$-values (Fig. 4c). In the Methods we show, that in the 1d-model, this phenomenology is due to the combination of two distinct effects: for long $\tau$ at melting only a few particles overcome the barrier by moving at nearly constant propulsion force, instead of diffusing (i), while the vast majority of particles are trapped and their fluctuations are "damped" upon increasing $\tau$ (ii), which is a well-known effect in active harmonic oscillators[41]. With these assumptions, we obtained a simple theoretical description of the active barrier-crossing which fits well the data in Fig. 4c, both for the 1d model and 2d simulations. In light of this model, we interpret the lowering of the melting Lindemann parameter in the active case as the result of individual persistent hopping events.

## Discussion
In conclusion, we used light-activated bacteria swimming through a colloidal crystal with tunable magnetic interactions to study how active fluctuations excite relaxation modes in a crystal and provide a non-equilibrium route to melting. We find that multiple active temperatures coexist in an active crystal with short wavelength modes being "colder" than long wavelength ones. When persistence is very significant, this results in a higher melting temperature along the active route. Moreover, compared with the passive case, a significantly lower critical threshold for the Lindemann parameter is found in active melting. Our findings reveal novel aspects in the behavior of active crystals through a simple and controllable experimental system. Future studies can investigate whether this phenomenology is robust

for different types of active noise[42] and how activity affects defect dynamics. Moreover, the experimental system we presented could be used further to study active magnetic glasses and the physics of the active glass transition[21,22] by using polydisperse magnetic particles. On the other hand, earlier theoretical work[16], in accordance with our observations, shows that crystals made of active particles melt following qualitatively the KTHNY two-step melting scenario observed in equilibrium. However, it was shown that other effective temperatures can be introduced which deviate from each other as persistence increases. Understanding how these earlier definitions[43] are related to those used here would be an important step toward a complete comprehension of the dynamics and thermodynamics of active solids and pave the way for practical applications of light-responsive smart materials in real-world scenarios.

## Methods

### Microscopy and light projection
We use a custom inverted optical microscope equipped with a 20x magnification objective (Nikon) and a high-sensitivity CMOS camera (Hamamatsu Orca-Flash 4.0 V3). Green light of tunable intensity is projected into the sample using a green LED.

### Magnetic field
An external constant magnetic field perpendicular to the sample plane is generated by passing a constant current through a custom-made coil connected to a power amplifier. The magnitude of the magnetic field at different currents was calibrated using a Teslameter (Exonder, MS-GAUSSMETER HGM09).

### Sample preparation
For all the experiments we used the *E. coli* wild-type (tumbling) strain AB1157 transformed with a plasmid encoding the proteorhodopsin (PR), and depleted of the *unc* cluster that codes for $F_1 F_o$-ATPase as described in ref. 44. *E. coli* colonies from frozen stocks are grown overnight at 30°C on LB agar plates supplemented with ampicillin (100 μg/mL). A colony is picked and cultivated overnight at 30°C at 200 rpm in 10 mL of LB with ampicillin. The next day, the overnight culture is diluted 100 times into 5 mL of TB containing ampicillin, and grown at 30°C, 200 rpm for 4h. Then all-*trans-retinal* (20 μM) and L-arabinose (1 mM) are added to ensure expression and proper folding of PR in the membrane. The cells are collected after 1 hour of induction by centrifugation at 1300 g for 12 min in a 15 ml tube at room temperature. The supernatant is removed and the bacterial pellet is resuspended in 1 ml motility buffer, containing 0.01% of Tween® 20. The suspension is then transferred to a 1.5 ml tube and the cells are washed by centrifugation of 5 min three additional times. Finally, the cells are resuspended at the desired concentration and vortexed for 20 seconds. This medium allowed the cells to be motile without allowing growth or replication, so the concentration of the cells remained constant throughout the experiments.

The used magnetic colloids (Dynabeads M-450, Thermo Fisher Scientific) have spherical shape and a diameter of 4.5 μm. They are highly monodisperse (narrow size distribution, coefficient of variation < 1.5%) and are doped with ferrite (~20%), which gives them a density of $\rho = 1.6$ g/cm³, and a magnetic volume susceptibility $\chi = 0.4$[11,45]. These magnetic microspheres are composed of a dispersion of super-paramagnetic nanoparticle grains made from iron oxides ($Fe_3O_4$ or γ-$Fe_2O_3$), which are uniformly and randomly distributed within a spherical porous host matrix and separated enough to not interact and to show superparamagnetic behavior.

Magnetic particles are diluted 100 times and washed in a basic solution of water and 1% of Tris-Base (pH 9) and Tween® 20 (0.2%). They are then sonicated for 15 mins, collected with a magnet, and prepared at the desired concentration in water and Tween® 20 (0.2%).

Slides (76 mm × 26 mm) and cover glasses (24 mm × 24 mm) are cleaned with water and soap and then coated with Tween® 20: they are sandwiched with 8 μL of Tween® 20 in-between and left during 48h at 37 °C and then baked during 45 min at 60 °C.

To achieve the desired confinement approximately equal to the particle diameter ~ 4.5 μm, 2 μL of the resulting mixture containing both particles and bacteria ( ~1:1) is deposited onto the microscope slide. Subsequently, gentle pressure is applied to the cover glass until the solution fully wets its surface. The sample is then sealed with vaseline.

Oxygen in the sample is depleted by bacteria in a few minutes. Once this has happened, the bacteria swim only where there is green light, and their speed increases with the intensity of the light.

The field of view measures 666 μm × 666 μm, accommodating typically more than 1000 particles with an inter-particle separation ranging between 20-25 μm.

### Active magnetic crystal measurements
In the absence of a field, the particles have no net magnetic moment and perform simple thermal diffusion. When a field of strength $H$ is applied, their induced moment becomes $\mathbf{m} = \chi \mathbf{H}$ where $\chi$ is the particle's magnetic susceptibility[46]. The dipolar interaction potential between two dipoles $\mathbf{m}_i$ and $\mathbf{m}_j$ at a distance $r_{ij} = r_i - r_j$ is $U = -\mu_0/4\pi([3(\mathbf{m}_i \cdot \mathbf{r}_{ij})(\mathbf{m}_j \cdot \mathbf{r}_{ij})/r_{ij}^5] - (\mathbf{m}_i \cdot \mathbf{m}_j)/r_{ij}^3)$, where $\mu_0 = 1.26 \cdot 10^{-6} N/A^2$ is the vacuum magnetic permeability.

To assemble the magnetic crystal a constant and high magnitude magnetic field ($H \sim 10$mT) perpendicular to the sample plane. The system was then equilibrated for around 1h until a monocrystalline state was reached. Then the bacterial bath was activated by using green light and measurements at different green light intensities, magnetic field were made. For the measurements of the melting, an initial crystalline phase was assembled and then the parameters were changed and then left equilibrated. The measurements of the passive melting were made without bacteria but in the same buffer.

To calibrate the experimental parameters $D_T$, $D_A$ and $\tau$ we adopt the following procedure. Since the over-damped dynamics of the *i*-th colloids in the active crystal can be modelled with the following stochastic differential equation:

$$\dot{\mathbf{r}}_i = \mu \mathbf{F}_i + \boldsymbol{\eta}_i + \boldsymbol{\xi}_i \tag{10}$$

where $\mathbf{F}_i = \sum_{i\neq j} \mathbf{f}(r_{ij})$ is the force due to the magnetic dipolar interactions, i.e. $\mathbf{f}(r_{ij}) = -\nabla_{\mathbf{r}_i} U_M(r_{ij})$ with $r_{ij} = |\mathbf{r}_i - \mathbf{r}_j|$. Here $\boldsymbol{\xi}_i$ and $\boldsymbol{\eta}_i$ are the thermal and active noise terms, respectively, with $\langle \xi_i^\alpha(t)\xi_j^\beta(t) \rangle = 2D_T \delta_{\alpha\beta}\delta_{ij}\delta(t-t')$ and $\langle \eta_i^\alpha(t)\eta_j^\beta(t) \rangle = D_A \delta_{\alpha\beta}\delta_{ij} \exp(-|t-t'|/\tau)/\tau$, where the Greek indices refer to the Cartesian components.

From Eq. (10) we can obtain that the mean squared displacement in the absence of interactions ($\mathbf{F}_i = \mathbf{0}$) is:

$$\langle \Delta r_i^2(t) \rangle = 4D_T t + 4D_A[t - \tau(1 - e^{-t/\tau})] \tag{11}$$

For each experiment we calibrate the active diffusion coefficient $D_A$ and the persistence time of the active motion $\tau$ in the absence of a magnetic field for different green light intensities $I$. For that, we measure the mean-squared displacement $\langle \Delta r_i^2(t) \rangle$ for each sample at each $I$ and fit it to Eq. (11) (see Supplementary Fig. 1).

### Analysis of active and passive mode band structure
To study the relaxation modes of the active crystal (both in experiments and simulations) we proceed as in ref. 10 by first computing the Fourier-Transformed displacement field as $\mathbf{u_q} = N^{-1/2}\sum_i \mathbf{u}_i e^{i\mathbf{q}\cdot\langle\mathbf{r}_i\rangle}$ where the $\langle\mathbf{r}_i\rangle$ are the average (equilibrium) positions of the $N$ particles. Here $\mathbf{u_q}$ is computed only for the wavevectors $\mathbf{q}$ that lie in the first Brillouin zone[35]. The correlation matrix $\langle \mathbf{u_q^*}\mathbf{u_q} \rangle$ is then computed as an

average over all configurations and the $p_{\mathbf{q}}^s$ are extracted as the eigenvalues of this matrix. In the case of thermal (passive) systems the eigenvalues $k_{\mathbf{q}}^s$ can be computed from the $p_{\mathbf{q}}^s$ by using the energy equipartition theorem (3), i.e. $\mu k_{\mathbf{q}}^s = D_T/p_{\mathbf{q}}^s$, and those are used in analysis presented in Fig. 2. We recall that the $k_{\mathbf{q}}^s$ are the eigenvalues of the dynamical matrix $\mathbf{D}_{\mathbf{q}}$ (as defined in ref. 35) which is the Fourier-transform of the real-space matrix $\mathbf{D}(\langle\mathbf{r}_i\rangle - \langle\mathbf{r}_j\rangle)$. This is the matrix of the second derivatives of the total potential energy $\mathcal{U}$ evaluated in the particles equilibrium positions (i.e. at zero displacement):

$$D^{\alpha,\beta}(\langle\mathbf{r}_i\rangle - \langle\mathbf{r}_j\rangle) = \left.\frac{\partial^2 \mathcal{U}}{\partial u^\alpha(\langle\mathbf{r}_i\rangle)\partial u^\beta(\langle\mathbf{r}_j\rangle)}\right|_{\mathbf{u}=\mathbf{0}} \quad (12)$$

where $\alpha$ and $\beta$ represent the Cartesian components. To compute the theoretical eigenvalues $k_{\mathbf{q}}^s$ (shown in Fig. 2a and b) for the crystal with long-ranged magnetic dipolar interactions we proceed as in ref. 10. We first compute the matrix from Eq. (12) in the perfect triangular Bravais lattice, we then perform the Fourier-transform and we finally extract the eigenvalues of the resulting $\mathbf{D}_{\mathbf{q}}$.

## Simulations

We perform numerical simulations of $N = 2216$ self-propelled AOUP particles in 2d. An Ornstein-Uhlenbeck (OU) process provides a good representation for the active noise in these passive tracers in an active bacterial bath, since they have short time displacements that are nearly Gaussian distributed[39] and decorrelate exponentially with time (See Supplementary Fig. 6). The simulations do not incorporate hydrodynamic interactions because, in our experimental setup, they are effectively screened by the presence of two confining glass plates separated by about $5\mu m$. Furthermore, inter-particle distances are much larger than particle radii ($r/a \sim 10$), diminishing significantly the influence of hydrodynamic interactions. Particles positions evolve according to Eq. (10), using the Euler scheme with time step $dt = 10^{-3}s$. For a better comparison with experimental results we adjust the size of the simulation box to achieve a density approximately matching that observed in experiments, i.e. the resulting lattice spacing in the crystal is set to $a = 20\ \mu m$. The simulation box has size $L_x \times L_y$ where $L_x = a\sqrt{N}$ and $L_y = (\sqrt{3}/2)L_x$ (periodic boundary conditions apply). Furthermore, we fix the amplitude $A$ of the pair potential $u(r_{ij}) = A/r_{ij}^3$ to the value $A = \mu_0\chi^2 H^2/2\pi$ employed in experiments at high magnetic field ($H \approx 10$ mT). Finally, the potential is truncated at ten lattice units, which yields accurate results as detailed in Ref. 47. To investigate the Lindemann parameter and the orientational correlation function at different $T^*$ and $\tau$ values we average results over three independent runs for each state point. The thermal diffusivity is kept fixed at the value $D_T = 0.03\ \mu m^2/s$, i.e. the value estimated from experiments in the absence of activity and of the magnetic field, and only $D_A$ and $\tau$ are varied. The melting temperature, denoted as $T_m^*$, and the corresponding critical Lindemann parameter at melting, $\gamma_m$, are defined as follows: $T_m^*$ represents the temperature at which the Lindemann parameter curve starts to diverge, showing an inflection point after the plateau within the observation time. The value of the Lindemann parameter at this inflection point is identified as $\gamma_m$.

## 1D model

The equations of motion for one-single AOUP moving in a cosine potential are given by

$$\dot{x} = \mu f(x) + \eta \quad (13)$$

$$\dot{\eta} = -\eta/\tau + (D_A^{1/2}/\tau)\xi \quad (14)$$

where $f(x) = -\partial_x A\cos(2\pi x/L)$, $\tau$ is the relaxation time of the noise, $D_A$ is the active diffusivity and $\xi$ is a delta-correlated Gaussian noise

source, i.e. $\langle\xi(t)\xi(t')\rangle = 2\delta(t-t')$. Simulations of the model (13) and (14) are performed in a box extending between 0 and $L$ (periodic boundary conditions apply). In all simulations, we fix the parameters $\mu = 1$, $L = 1$, and $A = 2 \times 10^{-2}$ and averaged over $10^3$ particles. Eqs. (13) and (14)) are numerically integrated by using the Euler scheme with a time step $dt = 10^{-3}$ and for a maximum time interval of $3 \times 10^6$ steps (data are collected only after $10^6$ steps to ensure that the system reaches the stationary state). We perform simulations scans by varying systematically $D_A$ to observe the hopping transition at different $\tau$-values. This is identified by checking if a non-zero slope at long times is present in the mean-squared-displacement $\langle\Delta x^2(t)\rangle$ (see Fig. 4c). We denote the active diffusivity needed to observe the transition as $D_M$ and find that this is an increasing function of $\tau$. We also find that the plateau value of $\langle\Delta x^2(t)\rangle$ at the transition (indicated by $\langle\Delta x^2\rangle_m$) decreases appreciably as $\tau$ increases (Fig. 4c).

## 1D theory

To rationalize the results of the 1d model we first consider that at the transition only a very small fraction of particles actually escapes from the minimum of the cosine potential (located at $x = L/2$), while the vast majority of the particles fluctuate around the minimum. Moreover, it is clear that these rare barrier-crossing events happen very differently at short and large $\tau$. Indeed for $\tau \approx 0$, we expect the hopping particle to execute a random walk, reversing their active force $\eta$ many times, before the barrier is crossed. The transition for $\tau = 0$ is observed for values of $D_M^0$ such that the effective temperature is of the order of the potential barrier, i.e. $D_M^0/\mu \approx 2A$. Differently as $\tau \to \infty$ the barrier is crossed by rare particles going straight with high and constant $\eta$ such that $\eta = \eta_M \approx \mu f_{\max}$ (where $f_{\max} = 2\pi A/L$ is the maximum force that the cosine potential opposes to the particles). Since $\eta$ is Gaussian-distributed with variance $D_A/\tau$ we have $\eta_M \approx \alpha\sqrt{D_M/\tau} \approx \mu f_{\max}$ where $\alpha$ is a parameter specifying how rare are hopping particles, i.e. how many "standard deviations" they differ from the mean $\eta = 0$ (e.g. if $\alpha = 3$ less than 1% of the particles overcome the barrier at the transition). We thus have that in the long-$\tau$ limit $D_M$ grows linearly with $\tau$: $D_M \approx (\mu^2 f_{\max}^2 \alpha^{-2})\tau$. Interpolating between the long and short $\tau$ regimes we thus approximate: $D_M \approx D_M^0 + (\mu^2 f_{\max}^2 \alpha^{-2})\tau$. Equivalently, by introducing the adimensional temperature $T^* = D_A/(2\mu A)$, we have $T_M^* \approx (2\mu A)^{-1}[D_M^0 + (\mu^2 f_{\max}^2 \alpha^{-2})]\tau$. We fix $D_0$ in this formula to the diffusivity at the transition found in simulations with $\tau = 0$ and use it to fit the data in Fig. 4c with $\alpha$ as a free parameter. The linear fit interpolates well the data and we find $\alpha \approx 3.8$. Finally, to predict the $\tau$-dependence of $\langle\Delta x^2\rangle_m$, we recall the almost all particles are fluctuating around the minimum so that, linearizing Eq. (13) as in ref. 41, we get the plateau value: $\langle\Delta x^2\rangle_m = D_M/[\mu k(1+\mu k\tau)] = [D_M^0 + (\mu^2 f_{\max}^2 \alpha^{-2})\tau]/[\mu k(1+\mu k\tau)]$, where $k = 4\pi^2 A/L^2$ is the curvature of the potential minimum. This equation (now without any free parameters) fits well the data in Fig. 4c.

## Data availability

Source data for all figures, both in the main text and in the Supplementary Information, are provided with the paper as a Source Data file. All the raw data that support the findings of this study are available from the corresponding authors upon request. Source data are provided with this paper.

## Code availability

The codes are corresponding to this study are available from the corresponding authors upon request.

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

## Acknowledgements

This project has received funding from the European Union's Horizon 2020 research and innovation programme under the Marie Skłodowska-Curie grant agreement No. 101019795 (H.M.-C.). H.M.-C. also acknowledges funding from ADD SapiExcellence 2023 (000008_23_RS_ADD_-MASSANA_CID - ADD 2023 MASSANA-CID - DI LEONARDO, Sapienza University of Rome). N.G. and C.M. acknowledge financial support from the project "MOCA" funded by MUR PRIN2022 grant No. 2022HNW5YL. R.D.L acknowledges funding from the European Research Council under the ERC Grant Agreement No. 834615.

## Author contributions

H.M.-C., C.M., G.F. and R.D.L. designed the experiments. H.M.-C. performed experiments and analyzed data. C.M. and N.G. performed simulations and analyzed data. G.F. and H.M.-C. were responsible for the growth of bacterial strains. H.M.-C., C.M., N.G., G.F. and R.D.L. wrote the manuscript.

## Competing interests

The authors declare no competing interests.
