## [Peer Review File · Nature Communications]

REVIEWER COMMENTS

Reviewer #1 (Remarks to the Author):

The manuscript "Multiple temperatures and melting of a colloidal active crystal" by Massana-Cid et al. presents an experimental setup where it is possible to independently control the interaction (repulsive) energy between magnetic colloids and the activity to which they are subjected. In dimensionless form, this is equivalent to independent control of the de thermodynamic temperature and activity. The authors use this system for two purposes. First, they analyze how the different vibrational modes are excited to test whether equipartition is satisfied. Second, they consider the high temperature case to study the melting of the crystal due to activity.

The setup is indeed very interesting, the experiments are detailed, and the analysis of the results is correct. The results presented have the potential to make a significant contribution to the field. However, there are several important issues that make the manuscript, in its current form, not suitable for publication in Nature Communications.

1) The two parts of the manuscript seem quite unrelated and lack a clear message. Is it possible to use the knowledge of the wavelength dependence of T_A in the description of melting?

2) There are some important details of the experimental setup that are not given:

- Size of the field of view, both in microns and in typical particle-particle distance.
- The average number of particles in the field of view.
- How is the 2D confinement achieved? Are there fixed size separators? What is the size of the gap?
- What is the typical particle size? Are they spherical? Monodisperse?
- Is strain AB1157 a tumbler or a smooth swimmer?

3) Perhaps it is simply semantic, but I am not convinced that it is appropriate to call temperatures the normalized variances of the different vibrational modes. At equilibrium, the temperature obtained as a variance (e.g. the mean kinetic energy) can be used later to calculate other thermodynamic quantities or to give the distribution functions. The active temperature, depending on the wave vectors, can be used to predict other quantities? Otherwise it risks being just a definition. For example, the manuscript says "We test Eq. (6) ...". If T_A is a definition, it is not clear what the authors mean by test.

4) The activity of the bacterial bath is manipulated by changing the intensity of the green light I_g , which the authors say controls the speed of the bacteria. It is not clear why the authors characterize the different states of the bath only by the memory time τ . Also, why does τ change with light intensity? In the supplementary material it is important to provide a calibration curve of τ , D_A and $V_A = \sqrt{D_A/\tau}$ vs. I_g .

After Eq. (6), τ appears for the first time and there is no indication of how it was obtained and what it means.

5) Figure 3a shows the melting diagram as a function of U_M and D_A . It would be interesting to plot the results as a function of D_T and D_A (2D plot) and as a function of $D_T + D_A$ to see if the two temperatures play similar roles and a collapse is obtained.

6) The decrease in the critical value of γ_L and the proposed explanation that this is due to the existence of persistent particles could imply that the PDF of Δ_r has long tails. Is it possible to show these distributions, for example, when the activity changes? In the same line, is it possible to provide evidence that the mechanism found in 1D operates in the 2D case?

7) The KTHNY melting theory is not easy to observe because it requires large systems to achieve the desired scale separation between lattice size, average defect spacing, and system size. The plots shown in Fig. 3c do not provide substantial evidence that the three regimes for g_6 (constant, power-law decay, and exponential decay) exist.

8) In the Methods section, before Eq. (10), it says "Since the dynamics of the i -th colloid [...] can be modelled with ...". What is the evidence that the colloids are indeed described as driven by an OU noise?

Minor details:

- The caption of Fig. 3 refers to the panels as (a), (c), and (d) instead of (a), (b), and (c).

- In Figure 3c, the top panel uses a color bar to indicate the temperatures, while the bottom panel gives the temperatures explicitly. This difference in presentation makes it difficult to compare the panels. Probably the easiest way is to give the temperature values explicitly in both cases.

- In the section "1D theory" of the Methods, the notation of symbols with subscript M is changed to subscript m.

Reviewer #2 (Remarks to the Author):

In this manuscript, authors have reported on the effect of active fluctuations on both the vibrational modes of a magnetic colloidal crystal and the melting route. The experiments seem to be carefully carried out and the Brownian-like dynamics confirmed the main observations reported by the authors. Overall, the manuscript is well written, organized and the results are clearly discussed and presented. Therefore, I recommend the publication of the manuscript in Nature Communications. However, it would be interesting if authors address the following points before the acceptance of the manuscript:

1. Authors should explain in detail the derivation (and motivation) of the generalized Stokes-Einstein relation (Eq. (1)). In particular, the definition of the magnetic interaction

energy must be properly introduced. Although this kind of two-dimensional magnetic colloidal system has been largely studied by other authors, its main features should be here rephrased.

2. Authors should briefly discuss on the expected physical scenario if the bacteria concentration is lower or higher than the one used in the experiments.

3. Authors concluded that multiple active temperatures coexist in their active colloidal crystal, but I failed to understand if the results suggest that the KTHNY theory can also be used to understand the active melting scenario here reported or whether it should be extended. This point must be explicitly discussed.

Reviewer #3 (Remarks to the Author):

In the submitted manuscript, the authors study an experimental system consisting of colloids repelling through magnetic dipolar interactions in a bacterial bath. They explore the properties of these active crystals and the melting transition. They complement their study with 2D numerical simulations and a simple 1D model.

I think this experimental system is a great way to explore theoretical ideas on active crystals that have attracted a lot of attention recently. It offers (I believe) unprecedented control with the possibility of tuning both activity and interactions rather precisely (robotic systems like in Ref.6 also offer a lot of control but are much more limited in particle number). I believe the main result is

the shift of the melting temperature with activity which is convincingly shown. Some theoretical arguments are also given. The "two-step" scenario from KTHNY I believe is less clear (see comment below). All in all, I think this paper is a nice addition to the literature and that it can be published in Nature Comm. after the comments below have been addressed.

1) The existing literature is not properly cited (to a surprising extent I must say). I do believe that at least some of the papers on active crystals should be acknowledged. Here are some examples, but the list is far from exhaustive and to be completed by the authors

Briand, G., Schindler, M., & Dauchot, O. (2018). Spontaneously flowing crystal of self-propelled particles. *Physical review letters*, 120(20), 208001.

Tan, T. H., Mietke, A., Li, J., Chen, Y., Higinbotham, H., Foster, P. J., ... & Fakhri, N. (2022). Odd dynamics of living chiral crystals. *Nature*, 607(7918), 287-293.

van Zuiden, B. C., Paulose, J., Irvine, W. T., Bartolo, D., & Vitelli, V. (2016). Spatiotemporal order and emergent edge currents in active spinner materials. *Proceedings of the national academy of sciences*, 113(46), 12919-12924.

Importantly, recent papers with numerics on much larger scale than in the present manuscript have studied in detail the KTHNY scenario for active crystals and are not cited.

Shi, X. Q., Cheng, F., & Chaté, H. (2023). Extreme spontaneous deformations of active crystals. *Physical Review Letters*, 131(10), 108301.

Shi, X. Q., & Chaté, H. (2024). Effect of Persistent Noise on the XY Model and Two-Dimensional Crystals. arXiv preprint arXiv:2401.11175.

As far as I can see it is not in contradiction with the present manuscript. I think it should be properly acknowledged and cited, especially when discussing the KTHNY.

2) "this melting scenario still holds when melting occurs mainly via active fluctuations"

From the data, it is not very clear. The authors should explain clearly for each curve why they believe that it belongs to one phase or the other. Actually from Fig.3c, I think it would be more interesting to have larger experimental panels with proper fits to power law or exponential decay rather than the (approximate) comparison with the simulations.

3) Typos:

- end of page 3, " the prediction in Eq(6)" should be Eq.(5)
- fig 3: letters in caption do not match
- p5 "comprehensive Lindemann melting value"

Reviewer #4 (Remarks to the Author):

Review for “Multiple temperatures and melting of a colloidal active crystal” by Massana-Cid et al.

The authors report an analysis of melting of an active 2d colloidal crystal. The direct colloidal interactions are controlled by a magnetic field and the activity by bacteria. Both may be independently controlled — the direct interactions by the external field and the bacteria by light. At low levels of activity, the authors claim that melting follows KTNHY theory as demonstrated some time ago by the Maret Group. The new twist induced by activity, according to the authors, is that it very significantly changes the melting scenario. In particular, the authors claim that equipartition is broken and multiple temperatures emerge, and melting occurs at small values of the Lindemann parameter.

There can be little doubt that the work the authors have done is very interesting. They have performed well—controlled experiments to explore a new phenomenon. However, the analysis is poor and not suitable for publication. In short, this is because the “collective vibrations” that the authors’ base much of their analysis on don’t actually exist. They do not exist in colloidal systems because they are over damped due to the liquid solvent in which the particles are immersed.

A robust analysis of how thermal excitations in colloidal systems may be analyzed has been provided by the Yodh lab — PRL 105, 025501 (2010). This paper shows how one may map fluctuations of an overdamped colloidal system onto a “shadow” system with Newtonian dynamics — that does have vibrations.

Now the inclusion of activity as we know, changes the dynamics massively. What is indisputable, and yet oddly not mentioned by the authors, is that hydrodynamic interactions have a massive role to play in this system. I understand that active colloids can “look” more like Newtonian particles in their motion. But that doesn’t mean that they are, or that there are vibrations as we understand them.

I think the authors should take a careful look at the literature on hydrodynamics in active matter, in particular papers citing Marchetti et al Rev. Mod. Phys. 85, 1143 (2013) and indeed that excellent review, and consider seriously what their system is doing.

The simulations look like they could do with some further work. The famous 2d melting simulations of Bernard and Krauth (PRL2011, PRE2013) used towards a million particles, to obtain suitable lengthscales as is a well-known issue in 2d melting. 2000 particles in 2024 just doesn't really cut it. Especially when the dipolar interaction is cut, rather than an Ewald sum being used. Even on a laptop, LAMMPS can routinely run tens of thousands of particles with an Ewald sum these days...

More fundamentally, since the simulations are being used to compare dynamic quantities, why are hydrodynamic interactions not used? (Which are also available in LAMMPS).

Related to the simulations (and their small system size), rather larger ranges of r would be expected to make claims about the $g_6(r)$ decay regimes in Fig. 3c. I think that lines showing the different scaling in the different phases (crystal, hexatic, fluid) would help for the experiments. Given the relatively small size of the imaging region, I understand that long ranges for g_6 may be challenging. So a statement like "the behavior is consistent with KTHNY" would be appropriate, rather stating that KTHNY behavior was found as it is not possible to conclusively show this, given the small size of the imaged region.

For the simulations, larger system sizes, to $r=100$ at least, would allow some more convincing evidence of the behavior of the system.

Regarding the literature, the authors seem to have missed the work which is the closest, technically, to theirs, at least in terms of the analysis used. This is a series of recent papers by Klongvessa and Leocmach and coworkers (2 PRE and a PRL at least, all since 2019). Here similar analyses were carried out on a slightly more disordered system.

In short, the experiments have uncovered interesting melting behavior in 2D active dipolar colloids. The analysis leaves much to be desired, and contains basic errors that need to be

addressed. In the present manuscript, the simulations add rather little to the story, not least given the issues with the analysis that the authors have followed, but also due to the small system size used to tackle a problem (2D melting) which is very well known for its huge lengthscales. I suggest that the authors make a very serious new analysis of these nice experiments.

Authors' reply to Reviewers comments

We thank all the referees for their thorough reading of our manuscript and for the constructive suggestions and criticisms. Our point-by-point responses and the corresponding changes to the manuscript are described below.

Response to Reviewer #1

Reviewer: *The manuscript "Multiple temperatures and melting of a colloidal active crystal" by Massana-Cid et al. presents an experimental setup where it is possible to independently control the interaction (repulsive) energy between magnetic colloids and the activity to which they are subjected. In dimensionless form, this is equivalent to independent control of the de thermodynamic temperature and activity. The authors use this system for two purposes. First, they analyze how the different vibrational modes are excited to test whether equipartition is satisfied. Second, they consider the high temperature case to study the melting of the crystal due to activity.*

The setup is indeed very interesting, the experiments are detailed, and the analysis of the results is correct. The results presented have the potential to make a significant contribution to the field.

However, there are several important issues that make the manuscript, in its current form, not suitable for publication in Nature Communications.

Authors: We thank the Reviewer for the positive comments on our manuscript. We have considered all the raised issues and recommendations and improved the manuscript accordingly.

Reviewer: *1) The two parts of the manuscript seem quite unrelated and lack a clear message. Is it possible to use the knowledge of the wavelength dependence of T_A in the description of melting?*

Authors: This is a very important point that we probably failed to highlight properly in the original manuscript. Our observations on active melting indicate that structurally and at large scales, the system is seemingly not affected by the wavelength dependence but can just be described with the global temperature T^* , as we show in Fig 3c. This is in line with our harmonic analysis and the wavelength dependence of T_A , which indicates that the wavelength modes that are most affected by activity are the short ones, and it suggests that the differences between active and passive melting routes might lie in local parameters instead. Consequently, we measured the Lindeman parameter, which captures fluctuations between nearest neighbors, and it indeed showcases the difference between active and passive melting routes (Fig 4). To clarify this in the text and bridge the gap between descriptions, we add in the manuscript:

"We have shown through harmonic analysis that the short wavelength modes are particularly affected by active noise, exhibiting decreased fluctuations as though they were in equilibrium

at a lower effective temperature compared to the longer wavelengths. This observation suggests that the peculiar aspects of active melting may emerge in measurements of local quantities, such as the Lindemann parameter which captures fluctuations between neighbors.”

Reviewer: 2) *There are some important details of the experimental setup that are not given:*

- *Size of the field of view, both in microns and in typical particle-particle distance.*
- *The average number of particles in the field of view.*
- *How is the 2D confinement achieved? Are there fixed size separators? What is the size of the gap?*
- *What is the typical particle size? Are they spherical? Monodisperse?*
- *Is strain AB1157 a tumbler or a smooth swimmer?*

Authors: We thank the reviewer for noticing these missing experimental details and we incorporate the information covering all raised aspects in the manuscript. Specifically, we add a scale bar in all experimental snapshots of Figure 1, and add the following text in the Methods section:

“For all the experiments we used the E. coli wild-type (tumbling) strain AB1157 transformed with a plasmid encoding the proteorhodopsin (PR), and depleted of the unc cluster that codes for F1 Fo-ATPase as described in Ref. [41].”

“The used magnetic colloids (Dynabeads M-450, Thermo Fisher Scientific) have spherical shape and a diameter of 4.5 μm . They are highly monodisperse (narrow size distribution, coefficient of variation < 1.5%) and are doped with ferrite (~20%), which gives them a density of $\rho = 1.6 \text{ g/cm}^3$, and a magnetic volume susceptibility $\chi = 0.4$ [10,42]. These magnetic microspheres are composed of a dispersion of superparamagnetic nanoparticle grains made from iron oxides (Fe_3O_4 or $\gamma\text{-Fe}_2\text{O}_3$), which are uniformly and randomly distributed within a spherical porous host matrix and separated enough to not interact and to show superparamagnetic behavior.”

“To achieve the desired confinement approximately equal to the particle diameter ~ 5 μm , 2 μL of the resulting mixture containing both particles and bacteria (1:1) is deposited onto the microscope slide. Subsequently, gentle pressure is applied to the cover glass until the solution fully wets its surface. The sample is then sealed with vaseline.”

“The field of view measures 666 μm x 666 μm , accommodating typically more than 1000 particles with an inter-particle separation ranging between 20-25 μm .”

Reviewer: 3) *Perhaps it is simply semantic, but I am not convinced that it is appropriate to call temperatures the normalized variances of the different vibrational modes. At equilibrium, the temperature obtained as a variance (e.g. the mean kinetic energy) can be used later to calculate other thermodynamic quantities or to give the distribution functions. The active temperature, depending on the wave vectors, can be used to predict other quantities? Otherwise it risks being just a definition. For example, the manuscript says "We test Eq. (6) ...". If T_A is a definition, it is not clear what the authors mean by test.*

Authors: Here we use an empirical notion of temperature that has been fruitfully employed to describe a similar multi-temperature scenario found in aging glass systems. This connection is better discussed in the revised manuscript, where we have made the following additions, including a new reference:

“A similar notion of effective temperatures has been introduced for glassy systems [34], where multiple temperatures can be measured by “thermometers” that probe the system’s dynamics on different timescales. In the present case, the relaxation modes in the crystal play the role of thermometers that measure different effective temperatures, revealing the non-equilibrium nature of the active system.”

[34] Kurchan, J. In and out of equilibrium. *Nature* 433, 222–225 (2005)

Regarding the general applicability of this temperature for other thermodynamic quantities of this specific non-equilibrium system is certainly a question that deserves further investigation. Recent work [Caprini, Lorenzo, et al. "Entropions as collective excitations in active solids." *The Journal of Chemical Physics* 159.4 (2023).] has shown that the same mode specific temperatures control entropy production in active crystals. Unfortunately, we cannot independently measure entropy production in our experiments, but this is definitely something worth investigating in future work, perhaps using recently proposed strategies for quantifying irreversibility [Ro, Sunghan, et al. "Model-free measurement of local entropy production and extractable work in active matter." *Physical Review Letters* 129.22 (2022): 220601].

On the other hand, following the Reviewer's suggestion and noticing a typo in the Equation reference, we correct the phrasing to avoid confusion:

“Using Eq. (6) to extract T_A we test Eq. (5) in Fig. 2g for an experimental active crystal...”

Reviewer: 4) *The activity of the bacterial bath is manipulated by changing the intensity of the green light I_g , which the authors say controls the speed of the bacteria. It is not clear why the authors characterize the different states of the bath only by the memory time τ . Also, why does τ change with light intensity? In the supplementary material it is important to provide a calibration curve of τ , D_A and $V_A = \sqrt{D_A/\tau}$ vs. I_g . After Eq. (6), τ appears for the first time and there is no indication of how it was obtained and what it means.*

Authors: We thank the reviewer for pointing out this gap. Active noise is characterized by both amplitude and correlation time, but it is only when τ becomes comparable to other relevant time scales in the problem that non-equilibrium signatures become more prominent. This is why in Fig.2 we compare small τ vs large τ systems to illustrate deviations from equilibrium behavior. We characterized activity induced in the particles by the bacterial bath by both the parameters D_A and τ . We make it more clear by adding in the manuscript: “When we apply green light $I \neq 0$, the bacteria swim and push the particles so the amplitude of their fluctuations around their lattice position increases (Fig. 1c). This results in colloids acquiring a persistent motion with a characteristic time τ and an effective active diffusion coefficient D_A . The effect of the active bath on the colloids, influenced by factors such as bacteria concentration and light intensity I (speed), can be characterized by D_A and τ [23]. We calibrate these parameters for each experiment, in the absence of a magnetic field for various green light intensities (see Methods and Supplementary Figure 1).”

We also rephrase in the Methods, for clarity:

“For each experiment we calibrate the active diffusion coefficient D_A and the persistence time of the active motion τ in the absence of a magnetic field for different green light intensities I . For that, we measure the mean-squared displacement $\langle \Delta r_i^2(t) \rangle$ for each sample at each I and fit it to Eq. (11) (see Supplementary Figure 1)”

We also add a new Supplemental Figure 1 that illustrates this calibration of the parameters D_A and τ :

Reviewer: 5) Figure 3a shows the melting diagram as a function of U_M and D_A . It would be interesting to plot the results as a function of D_T and D_A (2D plot) and as a function of $D_T + D_A$ to see if the two temperatures play similar roles and a collapse is obtained.

Authors: The referee raises an interesting point. We do not directly control D_T but we can only independently tune H and D_A to achieve different values of T^* that includes both active and passive contributions (see Eq.1). In this way we can still look for a data collapse when plotting the order parameter as a function of T^* . Despite some experimental noise potentially arising from slight variations in system sizes or the tails for large activities (see response of next point and new Supplemental Figure 5), we indeed find a nice collapse in the data:

To reference this, we add in the main text:

“Furthermore, if we plot the bond order parameter Ψ_6 all data of the phase diagram as a function of T^* we see that all points fall approximately on the same curve (Supplemental Figure 2).”

Reviewer: 6) *The decrease in the critical value of γ_L and the proposed explanation that this is due to the existence of persistent particles could imply that the PDF of Δr has long tails. Is it possible to show these distributions, for example, when the activity changes?*

Authors: The decrease of the critical γ_L is also captured by the simple 1D model. In this case, it is possible to justify the tau-dependence of γ_L with schematic theoretical arguments. These arguments assume Gaussian tails, so we believe that long tails are not a fundamental ingredient of this phenomenology. That said, we do see long tails in the experiments, as now shown in the new Supplemental Figure 5, and suggest that these tails may be at the origin of the lower value of the critical γ_L observed in experiments when compared to simulations. This is now discussed in the main text:

“We see that in the experiments active melting occurs at lower plateaus for the experimental persistence time (0.1-1s) than in simulations (1s-10s). This difference may be caused by the latter not capturing the longer tails of the displacement probabilities $P(\Delta x)$ observed experimentally, which could potentially trigger earlier melting. Specifically, we observe wider tails in the distributions when the activity D_A increases (Supplemental Figure 3).”

This is now illustrated in the new Supplemental Figure 5:

Reviewer: *In the same line, is it possible to provide evidence that the mechanism found in 1D operates in the 2D case?*

Authors: We acknowledge that this aspect may not have been adequately clarified in the main text. Figures 4c show that our simple theoretical argument is able to reproduce the tau dependence of the melting temperature and critical γ_L that we observe both in the 1D model and the 2D simulations. We believe this provides the evidence in favor of the existence of a common mechanism operating in both 1D and 2D systems. This is now better highlighted in the revised manuscript:

“With these assumptions we derive a simple theoretical description for the active barrier-crossing which fits well the data in Fig. 4c, **both for the 1d model and 2d simulations.**”

Reviewer: 7) *The KTHNY melting theory is not easy to observe because it requires large systems to achieve the desired scale separation between lattice size, average defect spacing, and system size. The plots shown in Fig. 3c do not provide substantial evidence that the three regimes for g_6 (constant, power-law decay, and exponential decay) exist.*

Authors: We agree with the reviewer that to confirm that the KTHNY theory is tricky to observe. So, as also Reviewer 4 suggested, we change our statement from “*this melting scenario still holds when melting occurs mainly via active fluctuations*” to “*We find that when melting occurs mainly via active fluctuations, it is still consistent with KTHNY*”. We also add fits to the data to power law and exponential decay (see new Figure 3c), and perform simulations for bigger systems (16000 particles, See new Supplemental Figure 3 and 4). In these extended simulations we still observed the 2-step melting scenario, even for larger persistence time. We now add in the main text:

“Furthermore, we still observe the 2-step melting scenario for larger τ and larger systems (Supplementary Figures 3-4).”

Reviewer: 8) In the Methods section, before Eq. (10), it says “Since the dynamics of the i -th colloid [...] can be modelled with ...”. What is the evidence that the colloids are indeed described as driven by an OU noise?

Authors: It was previously demonstrated that passive particles in an active bacterial bath (see new Ref. [35]) have short time displacements that are nearly Gaussian distributed and decorrelate exponentially with time. This suggests that an Ornstein–Uhlenbeck (OU) process provides a good representation for the active noise in these systems, like ours. To confirm this further, we calculate the fluctuations of freely diffusing colloids from their initial position $P(\Delta x)$ in our experiments of freely diffusing particles (dilute and without external magnetic field) in a bath under two different light intensities:

We indeed see that the distributions fit well a gaussian function, except for some long tails as also observed for confined colloids in the crystal and explained above in this response.

We add this as new Supplemental Figure 6 and the following text in the methods:

“An Ornstein–Uhlenbeck (OU) process provides a good representation for the active noise in these passive tracers in an active bacterial bath, since they have short time displacements that are nearly Gaussian distributed [35] and decorrelate exponentially with time (See Supplemental Figure 6)”.

We add the new citation:

[35] Maggi, C., Paoluzzi, M., Angelani, L. & Di Leonardo, R. Memory-less response and violation of the fluctuation-dissipation theorem in colloids suspended in an active bath. *Scientific reports* 7, 17588 (2017)

Reviewer: *Minor details:*

- The caption of Fig. 3 refers to the panels as (a), (c), and (d) instead of (a), (b), and (c).
- In Figure 3c, the top panel uses a color bar to indicate the temperatures, while the bottom panel gives the temperatures explicitly. This difference in presentation makes it difficult to compare the panels. Probably the easiest way is to give the temperature values explicitly in both cases.
- In the section "1D theory" of the Methods, the notation of symbols with subscript M is changed to subscript m .

Authors: We thank the reviewer for noticing these typos and minor corrections. We correct them and highlight all the changes in blue.

Response to Reviewer #2

Reviewer: *In this manuscript, authors have reported on the effect of active fluctuations on both the vibrational modes of a magnetic colloidal crystal and the melting route. The experiments seem to be carefully carried out and the Brownian-like dynamics confirmed the main observations reported by the authors. Overall, the manuscript is well written, organized and the results are clearly discussed and presented. Therefore, I recommend the publication of the manuscript in Nature Communications.*

Authors: We thank the referee for the very positive comments on our manuscript and for recommending publication in Nature Communications. We respond to all of the raised questions below.

Reviewer: *However, it would be interesting if authors address the following points before the acceptance of the manuscript:*

1. Authors should explain in detail the derivation (and motivation) of the generalized Stokes-Einstein relation (Eq. (1)). In particular, the definition of the magnetic interaction energy must be properly introduced. Although this kind of two-dimensional magnetic colloidal system has been largely studied by other authors, its main features should be here rephrased.

Authors: We agree with the reviewer that the phrasing might have been a bit confusing. We now describe more explicitly the generalization of the Stokes-Einstein relation and refer for a more detailed discussion to Ref. [30]. New text added before Eq. (1):

"In the limit of short persistence time τ , active particles with mobility μ behave like "hotter" equilibrium systems with a higher thermal energy $(D_A + D_T)/\mu$ obtained from a generalized Stokes-Einstein relation [30]. Dividing this by the magnetic interaction energy scale U_M we can introduce an adimensional global temperature T^* :"

[30] Bechinger, C. et al. Active particles in complex and crowded environments. *Reviews of Modern Physics* 88, 045006 (2016).

Reviewer: 2. *Authors should briefly discuss on the expected physical scenario if the bacteria concentration is lower or higher than the one used in the experiments.*

Authors: The reviewer makes an important point. The bacteria concentration highly affects the induced activity in the colloids [Valeriani, Chantal, et al. "Colloids in a bacterial bath: simulations and experiments." *Soft Matter* 7.11 (2011): 5228-5238.]. Our experiments are at a similar range of concentrations, but even then, for every experiment we calibrate the active character of the bath characterized by the parameters of active diffusion D_A and persistence time τ . We add a new Supplementary Figure 1 and the text:

"The effect of the active bath on the colloids, influenced by factors such as bacteria concentration and light intensity I (speed), can be characterized by DA and τ [23]. We calibrate these parameters for each experiment, in the absence of a magnetic field for various green light intensities (see Methods and Supplementary Figure 1)."

On the other hand, if the reviewer is referring to significantly larger bacterial concentrations, bacterial turbulence could appear. That would lead to a completely new regime and phenomena and would be an interesting future prospect.

Reviewer: 3. *Authors concluded that multiple active temperatures coexist in their active colloidal crystal, but I failed to understand if the results suggest that the KTHNY theory can also be used to understand the active melting scenario here reported or whether it should be extended. This point must be explicitly discussed.*

Authors: We believe that our experiments and simulations are consistent with the KTHNY scenario. This is not in contradiction with the multiple effective temperature picture. Indeed the harmonic analysis shows that fluctuations at long wavelengths are practically not affected by the persistence of the colored noise while the effect of activity is more evident at short wavelengths. Considering that KTHNY is essentially a coarse-grained theory of defects we believe that it cannot capture the peculiar small-scale physics of the active system but it still correctly reproduces (at least qualitatively) the two-step melting that we observe in experiment and in simulations. The harmonic analysis inspired us indeed to look at more local features that signal the melting transition such as the dynamic Lindemann parameter. To clarify this we now add in the manuscript:

"We have shown through harmonic analysis that the short wavelength modes are particularly affected by active noise, exhibiting decreased fluctuations as though they were in equilibrium at a lower effective temperature compared to the longer wavelengths. This observation suggests that the peculiar aspects of active melting may emerge in measurements of local quantities, such as the Lindemann parameter which captures fluctuations between neighbors."

Response to Reviewer #3

Reviewer: *In the submitted manuscript, the authors study an experimental system consisting of colloids repelling through magnetic dipolar interactions in a bacterial bath. They explore the properties of these active crystals and the melting transition. They complement their study with 2D numerical simulations and a simple 1D model.*

I think this experimental system is a great way to explore theoretical ideas on active crystals that have attracted a lot of attention recently. It offers (I believe) unprecedented control with the possibility of tuning both activity and interactions rather precisely (robotic systems like in Ref.6 also offer a lot of control but are much more limited in particle number). I believe the main result is the shift of the melting temperature with activity which is convincingly shown. Some theoretical arguments are also given. The "two-step" scenario from KTHNY I believe is less clear (see comment below). All in all, I think this paper is a nice addition to the literature and that it can be published in Nature Comm. after the comments below have been addressed.

Authors: We thank the Reviewer for the positive and constructive comments on our manuscript. We have addressed all the recommendations and improved the manuscript accordingly.

Reviewer: *1) The existing literature is not properly cited (to a surprising extent I must say). I do believe that at least some of the papers on active crystals should be acknowledged. Here are some examples, but the list is far from exhaustive and to be completed by the authors*

Briand, G., Schindler, M., & Dauchot, O. (2018). Spontaneously flowing crystal of self-propelled particles. Physical review letters, 120(20), 208001.

Tan, T. H., Mietke, A., Li, J., Chen, Y., Higinbotham, H., Foster, P. J., ... & Fakhri, N. (2022). Odd dynamics of living chiral crystals. Nature, 607(7918), 287-293.

van Zuiden, B. C., Paulose, J., Irvine, W. T., Bartolo, D., & Vitelli, V. (2016). Spatiotemporal order and emergent edge currents in active spinner materials. Proceedings of the national academy of sciences, 113(46), 12919-12924.

Importantly, recent papers with numerics on much larger scale than in the present manuscript have studied in detail the KTHNY scenario for active crystals and are not cited.

Shi, X. Q., Cheng, F., & Chaté, H. (2023). Extreme spontaneous deformations of active crystals. Physical Review Letters, 131(10), 108301.

Shi, X. Q., & Chaté, H. (2024). Effect of Persistent Noise on the XY Model and Two-Dimensional Crystals. arXiv preprint arXiv:2401.11175.

As far as I can see it is not in contradiction with the present manuscript. I think it should be properly acknowledged and cited, especially when discussing the KTHNY.

Authors: We thank the referee for pointing out that the bibliography was incomplete and some important references were missing. We have expanded the reference list with all papers suggested by the Reviewer plus some additional ones:

[11] Klamser, J. U., Kapfer, S. C. & Krauth, W. Thermodynamic phases in two-dimensional active matter. Nature communications 9, 5045 (2018).

[12] Paliwal, S. & Dijkstra, M. Role of topological defects in the two-stage melting and elastic behavior of active brownian particles. Physical Review Research 2, 012013 (2020).

[13] Pasupalak, A., Yan-Wei, L., Ni, R. & Ciamarra, M. P. Hexatic phase in a model of active biological tissues. *Soft matter* 16, 3914–3920 (2020).

[14] Digregorio, P., Levis, D., Cugliandolo, L. F., Gonnella, G. & Pagonabarraga, I. Unified analysis of topological defects in 2d systems of active and passive disks. *Soft Matter* 18, 566–591 (2022).

[15] Shi, X.-q., Cheng, F. & Chaté, H. Extreme spontaneous deformations of active crystals. *Physical Review Letters* 131, 108301 (2023).

[16] Shi, X.-q. & Chaté, H. Effect of persistent noise on the xy model and two-dimensional crystals. *arXiv preprint arXiv:2401.11175* (2024)

[20] Briand, G., Schindler, M. & Dauchot, O. Spontaneously flowing crystal of self-propelled particles. *Physical review letters* 120, 208001 (2018).

[21] Tan, T. H. et al. Odd dynamics of living chiral crystals. *Nature* 607, 287–293 (2022).

[22] Van Zuiden, B. C., Paulose, J., Irvine, W. T., Bartolo, D. & Vitelli, V. Spatiotemporal order and emergent edge currents in active spinner materials. *Proceedings of the national academy of sciences* 113, 12919–12924 (2016).

Moreover, we also expanded the introductory section to discuss the main findings in the added references:

“Whereas some numerical studies have shown that the two-step melting scenario of 2d crystals is qualitatively preserved when activity is introduced [11–14], more recent simulation work has pointed out that two different effective temperatures control the large scale elastic deformations of the crystal structure and the small-scale bond-order fluctuations [15,16].”

And in the conclusions:

“On the other hand, earlier theoretical work [15], in accordance with our observations, shows that crystals made of active particles melt following qualitatively the KTHNY two-step melting scenario observed in equilibrium. However, it was shown that other effective temperatures can be introduced which deviate from each other as persistence increases. Understanding how these earlier definitions are related to those used here would be an important step toward a complete comprehension of the dynamics and thermodynamics of active solids and pave the way for practical applications of light-responsive smart materials in real-world scenarios.”

Reviewer: 2) *“this melting scenario still holds when melting occurs mainly via active fluctuations”* From the data, it is not very clear. The authors should explain clearly for each curve why they believe that it belongs to one phase or the other. Actually from Fig.3c, I think it would be more interesting to have larger experimental panels with proper fits to power law or exponential decay rather than the (approximate) comparison with the simulations.

Authors: We modify Figure 3 and add the proper fits to power laws (dashed lines, correspond to crystalline or hexatic phase) and exponential decay (dotted lines, to isotropic):

Reviewer: 3) Typos:

- end of page 3, " the prediction in Eq(6)" should be Eq.(5)
- fig 3: letters in caption do not match
- p5 "comprehensive Lindemann melting value"

Authors: We thank the reviewer for noticing these typos and we fix them in the manuscript.

Response to Reviewer #4

Reviewer: *The authors report an analysis of melting of an active 2d colloidal crystal. The direct colloidal interactions are controlled by a magnetic field and the activity by bacteria. Both may be independently controlled — the direct interactions by the external field and the bacteria by light. At low levels of activity, the authors claim that melting follows KTNHY theory as demonstrated some time ago by the Maret Group. The new twist induced by activity, according to the authors, is that it very significantly changes the melting scenario. In particular, the authors claim that equipartition is broken and multiple temperatures emerge, and melting occurs at small values of the Lindemann parameter.*

There can be little doubt that the work the authors have done is very interesting. They have performed well—controlled experiments to explore a new phenomenon.

Authors: We thank the reviewer for appreciating the novel aspects of our work and for raising a number of questions and comments that helped us improve the manuscript as detailed in the responses below.

Reviewer: *However, the analysis is poor and not suitable for publication. In short, this is because the “collective vibrations” that the authors’ base much of their analysis on don’t actually exist. They do not exist in colloidal systems because they are over damped due to the liquid solvent in which the particles are immersed. A robust analysis of how thermal excitations in colloidal systems may be analyzed has been provided by the Yodh lab — PRL 105, 025501 (2010). This paper shows how one may map fluctuations of an overdamped colloidal system onto a “shadow” system with Newtonian dynamics — that does have vibrations.*

Now the inclusion of activity as we know, changes the dynamics massively. What is indisputable, and yet oddly not mentioned by the authors, is that hydrodynamic interactions have a massive role to play in this system. I understand that active colloids can “look” more like Newtonian particles in their motion. But that doesn’t mean that they are, or that there are vibrations as we understand them.

I think the authors should take a careful look at the literature on hydrodynamics in active matter, in particular papers citing Marchetti et al Rev. Mod. Phys. 85, 1143 (2013) and indeed that excellent review, and consider seriously what their system is doing.

Authors: We would like to clarify that all our analyses are indeed conducted in an overdamped regime, as demonstrated by Equation 10, where particle velocities are proportional to forces through a mobility coefficient. This is clearly not Newtonian dynamics and mass never enters our analysis. Normal modes in overdamped harmonic systems have been previously discussed along the same lines, although in equilibrium (see for example Reference [9] in the manuscript). To provide further clarification, we now explicitly state before Equation 10:

“Since the **over-damped** dynamics of the i -th colloids in the active crystal can be modeled with the following stochastic differential equation:...”

We acknowledge that the term 'vibration' may have led to some misunderstanding, even though it has often been used in the past to describe fluctuations in overdamped systems. In light of this, we have revised our terminology to only refer to “**relaxation modes**” and “**fluctuations**” instead.

Regarding hydrodynamic interactions, see the answer to the points below for a detailed explanation.

Reviewer: *The simulations look like they could do with some further work. The famous 2d melting simulations of Bernard and Krauth (PRL2011, PRE2013) used towards a million particles, to obtain suitable lengthscales as is a well—known issue in 2d melting. 2000 particles in 2024 just doesn’t really cut it. Especially when the dipolar interaction is cut, rather than an Ewald sum being used. Even on a laptop, LAMMPS can routinely run tens of thousands of particles with an Ewald sum these days...*

Authors: We had previously conducted simulations with ~2000 particles to match the order of magnitude of particles used in our experiments. However, in response to the Reviewer's suggestion and understanding the importance of scale, we have performed additional simulations with ~16,000 particles. The results of these simulations are presented in Supplementary Figures 3 and 4, and they showcase that our findings remain consistent across different particle numbers.

Reviewer: *More fundamentally, since the simulations are being used to compare dynamic quantities, why are hydrodynamic interactions not used? (Which are also available in LAMMPS).*

Authors: Hydrodynamic interactions have not been incorporated in the simulations due to the peculiar characteristics of our experiments:

- 1) Interparticle distances are much larger than particle radii ($r/a \sim 10$)
- 2) The crystal is confined in a 5 μm gap between two solid glass walls (no-slip). Coupling flows in this regime are known to decay faster than in the bulk $(a/r)^2$ [N. Liron, S. Mochon, J. Engineering Math., 10 287-303 (1976)].
- 3) Any residual hydrodynamic coupling between the beads would be very difficult to model because of the presence and movement of bacteria that strongly perturb the hydrodynamic field that couples colloids.

In such conditions, hydrodynamic interactions are significantly screened and therefore deemed negligible. We now emphasize this in the Methods section:

"The simulations do not incorporate hydrodynamic interactions because, in our experimental setup, they are effectively screened by the presence of two confining glass plates separated by about 5 μm . Furthermore, inter-particle distances are much larger than particle radii ($r/a \sim 10$), rendering the influence of hydrodynamic interactions negligible."

Reviewer: *Related to the simulations (and their small system size), rather larger ranges of r would be expected to make claims about the $g_6(r)$ decay regimes in Fig. 3c. I think that lines showing the different scaling in the different phases (crystal, hexatic, fluid) would help for the experiments. Given the relatively small size of the imaging region, I understand that long ranges for g_6 may be challenging. So a statement like "the behavior is consistent with KTHNY" would be appropriate, rather stating that KTHNY behavior was found as it is not possible to conclusively show this, given the small size of the imaged region. For the simulations, larger system sizes, to $r=100$ at least, would allow some more convincing evidence of the behavior of the system.*

Authors: We appreciate the reviewer's suggestions and acknowledge the importance of considering system size. We have adjusted the phrasing from 'this melting scenario still holds when melting occurs mainly via active fluctuations' to *'We find that when melting occurs mainly via active fluctuations, it is still consistent with KTHNY.'*

Furthermore, as suggested by the reviewer, we have included fits to the different phases and provided $g_6(r)$ data for larger simulated systems (new Supplementary Figures 3 and 4).

We conclude that the shift in the transition temperature for large tau seems to be not a simple (tau-dependent) finite-size effect, but it is probably a true temperature shift as observed by new Ref. [14]: Shi, X.-q., Cheng, F. & Chaté, H. Extreme spontaneous deformations of active crystals. Physical Review Letters 131, 108301 (2023).

We also add on the manuscript:

“Furthermore, we still observe the 2-step melting scenario for larger τ and larger systems (Supplementary Figures 3-4).”

Reviewer: *Regarding the literature, the authors seem to have missed the work which is the closest, technically, to theirs, at least in terms of the analysis used. This is a series of recent papers by Klongvessa and Leocmach and coworkers (2 PRE and a PRL at least, all since 2019). Here similar analyses were carried out on a slightly more disordered system.*

Authors: We thank the reviewer for pointing out these references, which we have now cited. However, these studies examine colloidal active solids that, because of short-range interactions, are formed at high packing fractions. Differently, in our case, long-range magnetic interactions are essential to have a crystal with a low packing fraction so that activity is not compromised by close neighbors and fluctuations are easily accessible. Moreover, the physics of the glass transition is very different from that of melting even in the passive case. For this reason, we think that it would be very interesting to study active magnetic glasses, perhaps by playing with the polydispersity of magnetic particles. This possibility is now suggested in the revised concluding section:

“Moreover, the experimental system we presented could be used further to study active magnetic glasses and the physics of the active glass transition [39,40] by using polydisperse magnetic particles.”

[39] Klongvessa, N., Ginot, F., Ybert, C., Cottin-Bizonne, C. & Leocmach, M. Active glass: Ergodicity breaking dramatically affects response to self-propulsion. *Physical review letters* 123, 248004 (2019).

[40] Klongvessa, N., Ginot, F., Ybert, C., Cottin-Bizonne, C. & Leocmach, M. Nonmonotonic behavior in dense assemblies of active colloids. *Physical Review E* 100, 062603 (2019)

Reviewer: *In short, the experiments have uncovered interesting melting behavior in 2D active dipolar colloids. The analysis leaves much to be desired, and contains basic errors that need to be addressed. In the present manuscript, the simulations add rather little to the story, not least given the issues with the analysis that the authors have followed, but also due to the small system size used to tackle a problem (2D melting) which is very well known for its huge lengthscales. I suggest that the authors make a very serious new analysis of these nice experiments.*

Authors: We thank the reviewer for his constructive comments. The basic error they refer to was just a misunderstanding that we hope to have cleared up. We have also taken up the suggestion to increase the size of the system hopefully getting more robust and convincing results from the simulations.

REVIEWER COMMENTS

Reviewer #1 (Remarks to the Author):

In the revised version of the manuscript, the authors have successfully answered the questions I have made. It is a very interesting work, which will stimulate discussions in the community. As such, I recommend the publication of the manuscript in Nat. Comm.

There is one issue that I would like to be considered for the final version. Both in experiments and in simulations (even for the larger system) the regime of power-law decay of Ψ_6 is not completely clear, with a dynamic range for distances that does not allow to discriminate convincingly between different decaying laws. I suggest to moderate the claim that the results are consistent with KTNHY. This modification does not alter the value of the manuscript, which, again I recommend for publication.

Reviewer #2 (Remarks to the Author):

Authors have correctly addressed all the points raised in my previous report. Also, authors have considered and answered all the concerns from the other reviewers. In particular, I find the manuscript of relevance for the community of active matter. I therefore recommend the publication of the manuscript in Nature Communications.

Reviewer #3 (Remarks to the Author):

Dear editor, my comments have been addressed and I can thus recommend publication.

Reviewer #4 (Remarks to the Author):

Second round of reviewing for “Multiple temperatures and melting of a colloidal active crystal” by Massana-Cid et al.

The authors have made some limited efforts to address the comments raised in the first round of reviewing. However, much remains to be done.

Reading the revised manuscript, one still has the impression that the authors seem to imagine that their colloidal particles are vibrating, even though, inconsistently, they state clearly in the response that the particles are overdamped.

The conclusions have not been altered at all — and in the revised manuscript, we still read of these “vibrating” particles.

And we still read of “modes”. I don’t really understand what the authors mean by “relaxation modes” — please explain.

Now a mode in the simple sense is defined as “a pattern of motion in which all parts of the system move sinusoidally with the same frequency and with a fixed phase relation”. This would be consistent with the authors’ use of the dynamical matrix — were they to have a Newtonian system, which they do not. What exactly are these modes that the authors see? And why do they seem to pursue a treatment which is not consistent with their own statements about the nature of the system.

The authors need to make a proper re-write of the manuscript. For a general audience like that of Nature Communications, they need to be clear about their methods — they certainly are not at the moment. They need to very clear about the following:

How is the dynamical matrix calculated? Is it calculated following the method of Chen et al PRL 105, 025501 (2010). This was discussed in the first round of reviewing, but seems to have been largely ignored by the authors. The point stands, though, what do the authors mean by “modes” in a system that doesn’t vibrate — by their own statement, but in any case, colloidal system are well known to be overdamped.

The only meaningful treatment that I am aware of is that of Chen et al (2010). That is to say, one can obtain the dynamical matrix of an equivalent Newtonian system. For an equilibrium colloidal system. I suppose that in some non—rigorous way one can claim something similar for an active system. But the author its be clear about what they are doing. And why they think it is OK.

At the moment, we read in the text around Eqs. 2 and 3 that the dynamical matrix can be determined from positions of the particles.....This is appropriate for a Newtonian system, but this is not what the authors are dealing with. Please get the story right, so that it is spelled out exactly what is done, in the context of the Newton showdown system (see Chen et al (2010)).

The reader needs to understand that, as an overdamped system, there are no vibrations in colloids. And that is far from the case in the present manuscript. And why is Chen et al not even cited?

Other points

Sweeping statements like HI are negligible quite simply have no place here. I can accept that HI are screened to some extent, but to cite a paper for 1976, just disregard the massive leaps that have been made in our understanding of HI since then. It's OK not to include HI in the simulation here, it is understood that properly including HI is a serious undertaking, and this isn't expected for a work whose main importance is the experiments.

Please take the time to look at the manuscripts by Klongvessa and Leocmach mentioned in the first round of reviewing. Even the most cursory glance at their data shows that the systems they study are in fact highly ordered. We know that the glass transition is not the same as crystallization. But the system those authors studied and publish on concerns rather crystalline state points.

Other papers in addition to those mentioned by reviewer (2) that really should be cited include the seminal work on effective temperatures with active colloids, which underlies so much of this manuscript that it is very surprising to see it not cited: Palacci et al PRL 105, 088304 (2010).

Authors' reply to Reviewers comments

We thank all the reviewers for their thorough review and valuable feedback on our manuscript. We are pleased that Reviewers #1,#2 and #3 find our revised manuscript suitable for publication in its current form. We have carefully considered the comments and suggestions provided by Reviewer #4. While we believe there may have been some misunderstandings regarding the analysis, we have made several revisions to improve the clarity of our manuscript. Our point-by-point responses and the corresponding changes to the manuscript are described below.

Response to Reviewer #1

Reviewer: *In the revised version of the manuscript, the authors have successfully answered the questions I have made. It is a very interesting work, which will stimulate discussions in the community. As such, I recommend the publication of the manuscript in Nat. Comm.*

Authors: We appreciate the reviewer's positive feedback on our manuscript and their recommendation for publication.

Reviewer: *There is one issue that I would like to be considered for the final version. Both in experiments and in simulations (even for the larger system) the regime of power-law decay of Ψ_6 is not completely clear, with a dynamic range for distances that does not allow to discriminate convincingly between different decaying laws. I suggest to moderate the claim that the results are consistent with KTNHY. This modification does not alter the value of the manuscript, which, again I recommend for publication.*

Authors: Recognizing the potential challenges in observing discrepancies with the KTNHY theory in active melting, we have adjusted our claims in the manuscript to reflect this consideration. In particular, we change the following sentences:

We find that in active melting, at large scales, ~~the system follows qualitatively~~ **the system appears to qualitatively follow** the two-step phase transition predicted by KTNHY theory...

~~We find that when melting occurs mainly via active fluctuations, it is still consistent with KTNHY~~ **When melting occurs mainly via active fluctuations, we do not observe any discernible discrepancies with KTNHY...**

Response to Reviewer #2

Reviewer: *Authors have correctly addressed all the points raised in my previous report. Also, authors have considered and answered all the concerns from the other reviewers. In particular, I find the manuscript of relevance for the community of active matter. I therefore recommend the publication of the manuscript in Nature Communications.*

Authors: We thank the Reviewer for the positive comments on our manuscript and for recommending publication.

Response to Reviewer #3

Reviewer: *Dear editor, my comments have been addressed and I can thus recommend publication.*

Authors: We appreciate the reviewer's positive feedback and recommendation for publication.

Response to Reviewer #4

Reviewer: *Second round of reviewing for "Multiple temperatures and melting of a colloidal active crystal" by Massana-Cid et al.*

The authors have made some limited efforts to address the comments raised in the first round of reviewing. However, much remains to be done. Reading the revised manuscript, one still has the impression that the authors seem to imagine that their colloidal particles are vibrating, even though, inconsistently, they state clearly in the response that the particles are overdamped. The conclusions have not been altered at all — and in the revised manuscript, we still read of these "vibrating" particles. And we still read of "modes". I don't really understand what the authors mean by "relaxation modes" — please explain.

Now a mode in the simple sense is defined as "a pattern of motion in which all parts of the system move sinusoidally with the same frequency and with a fixed phase relation". This would be consistent with the authors' use of the dynamical matrix — were they to have a Newtonian system, which they do not. What exactly are these modes that the authors see? And why do they seem to pursue a treatment which is not consistent with their own statements about the nature of the system.

Authors: We appreciate the reviewer's observation. We acknowledge that we missed one instance of the term "vibrational modes" in the conclusions when updating it to "relaxation mode" in the last revision to avoid misinterpretations in the analysis. This oversight has been rectified, and we thank the reviewer for bringing this to our attention. Having said that, we would like to stress once again that the word "modes" in our manuscript never refers to the oscillatory dynamics of phonons propagating in the crystal. In this overdamped system, indeed, the colloids can hardly oscillate due to solvent friction, and the "vibration modes" of Newtonian dynamics are now replaced by "relaxation modes". When one of these modes is excited in the crystal it exponentially relaxes to zero amplitude rather than oscillating as in a Newtonian system. The concept of "relaxation modes" has been widely used in the literature of colloidal crystals, such as in the seminal work by Keim, Maret, et al. Physical Review Letters 92, 2004, where they state:

"Our results demonstrate that a colloidal crystal can be seen as a bead-spring lattice immersed in a viscous fluid. A normal vibration mode then transforms into a "normal

relaxation mode” and the motion of a particle is to be understood as superposition of these “normal relaxation modes”. A time-dependent analysis of our data will allow one to study the relaxation process of these normal modes.”

Furthermore, in an earlier study from Ohshima et al. (The Journal of Chemical Physics 114, 2001) they state:

“As a result of the viscous media, traveling phonon modes are transformed into relaxation modes, and the motion of a particle is comprehended as a superposition of these relaxation modes.”

The Reviewer also seems to imply that the dynamical matrix would only be “consistent” within a Newtonian system displaying “sinusoidal motion”. Following classic solid state textbooks (i.e. Ashcroft-Mermin eq. 22.48) the dynamical matrix is just a measure of potential energy curvatures obtained by second derivatives around the lattice equilibrium position. It only describes the stiffness of the potential around equilibrium and it does not depend on masses. Therefore, even in practically massless systems (overdamped) the dynamical matrix is perfectly well defined and, rather than determining phonon frequencies, it defines a spectrum of relaxation rates. For this reason, the dynamical matrix has been a very useful concept to describe colloidal crystals (see for example eq. 1 in Keim et al, Phys Rev Lett 2004). In our manuscript, we define (and measure, in the equilibrium case) the dynamical matrix in exactly the same way and consider the case of an active system with active fluctuations.

With the hope to make this point more clear and unambiguous we have added a new sentence to the main text and included a new reference:

“We must importantly note that here we are not discussing phonon vibrational modes. Since the crystal is immersed in a viscous fluid, the colloids' motion is over-damped, and the trajectory of a particle is composed of a superposition of normal relaxation modes [36].”

[36] Ohshima, Y. N. & Nishio, I. Colloidal crystal: bead–spring lattice immersed in viscous media. The Journal of Chemical Physics 114, 8649–8658 (2001)

Reviewer: *The authors need to make a proper re-write of the manuscript. For a general audience like that of Nature Communications, they need to be clear about their methods — they certainly are not at the moment. They need to very clear about the following:*

How is the dynamical matrix calculated? Is it calculated following the method of Chen et al PRL 105, 025501 (2010). This was discussed in the first round of reviewing, but seems to have been largely ignored by the authors. The point stands, though, what do the authors mean by “modes” in a system that doesn't vibrate — by their own statement, but in any case, colloidal system are well known to be overdamped. The only meaningful treatment that I am aware of is that of Chen et al (2010). That is to say, one can obtain the dynamical matrix of an equivalent Newtonian system. For an equilibrium colloidal system. I suppose that in some non—rigorous way one can claim something similar for an active system. But the author its be clear about what they are doing. And why they think it is OK. At the moment, we read in the text around Eqs. 2 and 3 that the dynamical matrix can be determined from positions of the particles.....This is appropriate for a Newtonian system, but this is not what the authors are dealing with. Please get the story right, so that it is spelled out exactly what is done, in the context of the Newton showdown system (see Chen et al (2010)). The reader needs to

understand that, as an overdamped system, there are no vibrations in colloids. And that is far from the case in the present manuscript. And why is Chen et al not even cited?

Authors: We realize that the entire misunderstanding here stems from the Reviewer's belief that modes imply oscillations. We would like to clarify that, more generally, modes are just a convenient set of new coordinates that evolve independently, either oscillating (in Newtonian systems) or exponentially relaxing (in overdamped colloidal systems). Our use of the term aligns with a large body of literature (see also "Modes of motion of a colloidal crystal" by Hoppenbrouwers et al. Physical Review Letters 80 1998) and was later used by Keim, Maret et al. (Ref. 10 of the manuscript), among many others (Baumgartl et al. Soft Matter 4, 2008; Ueno et al. AIP Conf. Proc. 982, 2008; Mendoza et al. , Advances in colloid and interface science 206, 2014; Di Leonardo et al. Phys Rev E 76, 2007, E. Guzmán et al. Advances in Colloid and Interface Science, 2022). Based on such a solid background, we do not feel that a major rewrite is necessary to make the paper readable to a wide audience, which could be confused if we do not align with previous literature.

We now add a new reference to the methods when first mentioning the dynamical matrix ("Introducing the Fourier-transformed dynamical matrix [35] D_q (see Methods),...") and expand the explanation in the Methods of how we calculate the dynamical matrix:

"We recall that the k_q^s are the eigenvalues of the dynamical matrix D_q (as defined in Ref. [35]) which is the Fourier-transform of the real-space matrix $D(\langle \mathbf{r}_i \rangle - \langle \mathbf{r}_j \rangle)$. This is the matrix of the second derivatives of the total potential energy U evaluated in the particles equilibrium positions (i.e. at zero displacement):

$$D^{\mu,\nu}(\langle \mathbf{r}_i \rangle - \langle \mathbf{r}_j \rangle) = \left. \frac{\partial^2 U}{\partial u^\mu(\langle \mathbf{r}_i \rangle) \partial u^\nu(\langle \mathbf{r}_j \rangle)} \right|_{\mathbf{u}=0}$$

To compute the theoretical eigenvalues k_q^s (shown in Fig. 2(a) and (b)) for the crystal with long-ranged magnetic dipolar interactions we proceed as in Ref. [10]. We first compute the matrix from Eq. (12) in the perfect triangular Bravais lattice, we then perform the Fourier-transform and we finally extract the eigenvalues of the resulting D_q .

Regarding the connections to the Chen et al. paper, we believe that the main criticism of the Reviewer arises again from the assumption that "modes" imply underdamped vibrations. We hope that we have made it clear that in overdamped colloidal systems we can still define modes, even if their dynamics is only an exponential relaxation. On the other hand, Chen et al. study the vibrational dynamics of a "fictitious" shadow system with the same dynamical matrix as the colloidal one, but where there is no viscous damping. The reason for their approach is to establish a connection and study the vibrational spectrum of atomic glasses. Again, our system is strongly overdamped, no vibrations and therefore no need to consider a shadow, underdamped, system. Nevertheless, we now add the citation in the introduction of our manuscript to acknowledge their contribution to the field of colloidal model systems:

"They have been used to elucidate very debated issues connected to the nature of melting

transition in two dimensions [1], the dynamical arrest in glass transition [2], and the vibrational excitations in disordered solids [3].”

[3] Chen, K. et al. Low-frequency vibrations of soft colloidal glasses. *Physical review letters* 105, 025501 (2010)

Reviewer: *Other points: Sweeping statements like HI are negligible quite simply have no place here. I can accept that HI are screened to some extent, but to cite a paper for 1976, just disregard the massive leaps that have been made in our understanding of HI since then. It's OK not to include HI in the simulation here, it is understood that properly including HI is a serious undertaking, and this isn't expected for a work whose main importance is the experiments.*

Authors: We have now revised our statement to the following:

“The simulations do not incorporate hydrodynamic interactions because, in our experimental setup, they are effectively screened by the presence of two confining glass plates separated by about 5 μ m. Furthermore, inter-particle distances are much larger than particle radii ($r/a \sim 10$), ~~rendering the influence of hydrodynamic interactions negligible~~ diminishing significantly the influence of hydrodynamic interactions.”

Reviewer: *Please take the time to look at the manuscripts by Klongvessa and Leocmach mentioned in the first round of reviewing. Even the most cursory glance at their data shows that the systems they study are in fact highly ordered. We know that the glass transition is not the same as crystallization. But the system those authors studied and publish on concerns rather crystalline state points.*

Authors: We do agree that those are interesting and relevant studies and that is why we added the citation in the previous revision of the manuscript as per the Reviewer's suggestion. Now we also add the references to the introduction:

“While self-propelling synthetic active particles can self-organize into dynamic crystals when colliding [18–20], or into polycrystals when sedimenting [21,22],...

Reviewer: *Other papers in addition to those mentioned by reviewer (2) that really should be cited include the seminal work on effective temperatures with active colloids, which underlies so much of this manuscript that it is very surprising to see it not cited: Palacci et al PRL 105, 088304 (2010).*

Authors: That is indeed an interesting study about the applicability of thermodynamic concepts to active systems, in this case of a different system of sedimenting active colloids. We thank the reviewer for making us note this reference and add it to the manuscript:

[43] Palacci, J., Cottin-Bizonne, C., Ybert, C. & Bocquet, L. Sedimentation and effective temperature of active colloidal suspensions. *Physical Review Letters* 105, 088304 (2010)

REVIEWERS' COMMENTS

Reviewer #4 (Remarks to the Author):

Third round of reviewing for “Multiple temperatures and melting of a colloidal active crystal” by Massana-Cid et al.

The authors have now made a satisfactory job of clarifying the dynamical issues with their manuscript. They have also addressed the other points raised in the previous round of reviewing and I am now happy to recommend publication in Nature Communications.